# Escaping Whack-a-Mole: Optimizing Documentation as Repo-Specific Playbooks for Coding Agents

**Yutong Cheng** [1] [*]  **Haifeng Chen** [2]  **Wenchao Yu** [2]  **Xujiang Zhao** [2]  **Peng Gao** [1]  **Wei Cheng** [2]

## Abstract

As large language models increasingly function as autonomous coding agents, code documentation should be designed not for human readability, but for agent executability — serving as repo-specific playbooks that specify precise behaviors agents can follow. We formulate agent-oriented documentation generation as a black-box optimization problem over the documentation space, where quality is defined solely by downstream code correctness. A central challenge for conventional LLM refinement methods is *output coupling*—program entities are interdependent, and refining the documentation of one entity can invalidate its callers, resulting in a persistent *whack-a-mole* phenomenon during inference-time scaling. We propose DOCSEARCH, a dependency-guided bi-level search framework that systematically exploits test-time feedback. The outer level conducts a priority search over the program-entity dependency DAG, enforcing a callee-before-caller refinement order to prevent downstream interference. The inner level performs a beam search over documentation refinements, using diversified error message sampling from self-generated unit tests to better exploit diagnostic signals and escape local optima. On DevEval+, DOCSEARCH achieves 90.7% solve rate with GPT-4o, outperforming the strongest baseline by 32.6%. Cross-language experiments further demonstrate that optimized documentation transfers effectively to different target programming languages. Code is available at https://github.com/ccsnow127/docsearch.

---

[*]Work done during an internship at NEC Laboratories America. [1]Department of Computer Science, Virginia Tech, Blacksburg, VA, USA [2]NEC Laboratories America, Princeton, NJ, USA. Correspondence to: Wei Cheng <weicheng@nec-labs.com>.

*Proceedings of the 43rd International Conference on Machine Learning*, Seoul, South Korea. PMLR 306, 2026. Copyright 2026 by the author(s).

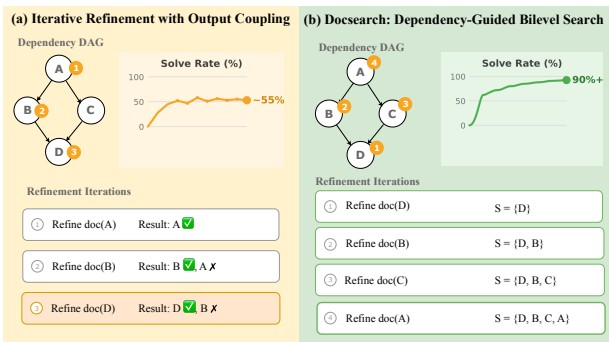

*Figure 1.* **Iterative refinement vs. DOCSEARCH.** (a) Output coupling induces a whack-a-mole effect: fixing one entity breaks another, causing the solve rate to oscillate around 55%. (b) DOCSEARCH processes entities in reverse topological order under a worthy condition, ensuring monotonic progress beyond 90%.

## 1. Introduction

Documentation bridges human intent and machine-generated code. As large language models become capable of synthesizing complex software modules, a fundamental shift occurs: documentation is increasingly consumed by *coding agents*, not just human developers. Repository-level code generation (Zhang et al., 2023a; Li et al., 2025), legacy modernization, and API reimplementation all rely on agents interpreting documentation to produce code. This shift demands a new perspective: documentation should serve as a repo-specific playbook for coding agents — an executable behavioral specification tailored to a particular codebase, rather than human-oriented prose. We formalize this task as *agent-oriented documentation generation*. Unlike human-oriented documentation, where quality can be assessed through subjective metrics like comprehensiveness, helpfulness, and clarity (Luo et al., 2024; Yang et al., 2025), agent-oriented documentation is evaluated by the *posterior* quality of code generated by an LLM or agent conditioned solely on the documentation, where correctness of the resulting implementation is the only reliable signal. This raises a fundamental question: how can such documentation be optimized? The space is vast and unstructured, with effectively infinite natural-language formulations and no intrinsic metric to guide search.

We formulate agent-oriented documentation generation as a

*black-box optimization problem* and address it using *LLM-based search*. The formulation arises naturally from two observations. First, test execution yields a well-defined, computable objective: candidate documentation is evaluated by generating code and measuring test pass rates against unit tests self-generated by the LLM from the original code, forming a closed-loop self-improvement process. Second, test failures contain rich diagnostic signals beyond binary pass/fail—error messages reveal type mismatches, assertion failures expose incorrect outputs, and stack traces indicate where execution diverged. This information directly reveals the agent's *knowledge gap*: what it misunderstood that caused incorrect generation. Together, these define a search structure: documentation states, test-driven transitions guided by failure analysis, and an objective function computed via posterior feedback. The optimization task is then to navigate this space, letting errors guide refinement toward documentation that enables correct code synthesis.

For this black-box optimization problem, a natural way is to employ scaling inference solutions (Brown et al., 2024; Gandhi et al., 2024). The most common approaches iteratively refine documentation using downstream test feedback, either via sequential (i.e., line-search) refinement (Shinn et al., 2023; Madaan et al., 2023), or by generating multiple documentation candidates and selecting the best using test-time evaluation—commonly known as *best-of-N* sampling (**?**Lightman et al., 2024). Alternatively, feedback can be exploited to revise solutions in a tree-structured manner (Wan et al., 2024; Chen et al., 2024a). However, incorporating code generation into the optimization loop introduces a challenge absent from typical optimization problems: **output coupling** (as illustrated in Figure 1a). Generated code entities exhibit call dependencies—when entity $A$ invokes entity $B$, the correctness of $A$'s code depends on $B$'s actual implementation, not merely its documentation. Refining $B$'s documentation may yield a different (though correct) implementation, breaking $A$ even when $A$'s documentation is unchanged. This coupling produces the **whack-a-mole phenomenon**: improving one entity breaks another; fixing that breaks a third; the search oscillates without convergence. We prove that greedy optimization achieves arbitrarily poor approximation ratios under output coupling (Theorem 1).

We propose DOCSEARCH, a dependency-guided *bi-level search* framework that navigates the documentation space while avoiding the *whack-a-mole* trap (Figure 1b). ❶ At the outer level, a **priority search** on the entity dependency DAG (Directed Acyclic Graph) selects *which entity* to refine first via reverse topological traversal of call dependencies, ensuring callees are refined before callers. ❷ At the inner level, a **beam tree search** explores *how to refine* the selected entity, with each beam branch leveraging *diversified sampling of error messages* from LLM self-generated unit tests. This not only explores multiple refinement directions but also helps escape local optima by exposing heterogeneous failure signals. Refinement proceeds via a two-stage *diagnosis-prescription* process: failures are analyzed to identify root causes, and targeted documentation updates are generated to address them. A *worthy condition* couples both levels, committing refinements only when they resolve the target without regressing previously solved entities, guaranteeing monotonic progress toward a globally correct solution.

To sum up, our contributions are as follows:

- We reframe documentation as repo-specific playbooks for coding agents and formulate playbook optimization as a black-box search problem; we identify *output coupling* as the root cause of the whack-a-mole phenomenon in iterative refinement (§2).

- We propose DOCSEARCH, a bi-level search approach that escapes the *whack-a-mole* trap via reverse topological traversal, error-driven refinement, beam search with diversified sampling and a worthy condition guaranteeing monotonic progress (§3).

- We validate DOCSEARCH on DEVEVAL+, demonstrating that it consistently outperforms strong baselines across LLMs, achieving a +32.6% improvement in ground-truth test pass rate over the leading baseline, exhibits monotonic progress while avoiding the whack-a-mole phenomenon, and enables effective cross-language transfer, validating optimized documentation as a language-agnostic, repo-specific playbook for coding agent. (§4).

## 2. Problem Formulation

We formalize agent-oriented documentation generation as a search problem and identify output coupling as its key distinction from standard optimization.

### 2.1. Agent-Oriented Documentation Generation

Consider a code repository containing $n$ entities $\mathcal{E} = \{e_1, \ldots, e_n\}$ (functions, classes, or methods) with call graph $G = (\mathcal{E}, E)$, where $(e_i, e_j) \in E$ indicates that $e_i$ invokes $e_j$. Each entity $e_i$ has associated documentation $d_i$ and a test suite $\mathcal{T}_i$ specifying correctness criteria.

**Definition 1** (Agent-Oriented Documentation Generation). *Given an agent (LLM) $\mathcal{G}$, documentation $\mathcal{D} = (d_1, \ldots, d_n)$ produces code $\mathcal{C} = (c_1, \ldots, c_n)$ where $c_i = \mathcal{G}(\mathcal{D}, e_i)$. The quality of documentation for entity $e_i$ is measured by the* pass rate:

$$\phi_i(\mathcal{D}) = \frac{1}{|\mathcal{T}_i|} \sum_{t \in \mathcal{T}_i} \mathbb{1}[\text{PASS}(c_i, t)], \tag{1}$$

*where $\mathbb{1}[\cdot]$ is the indicator function and $\text{PASS}(c_i, t)$ is true iff code $c_i$ passes test $t$. An entity is* solved *if $\phi_i(\mathcal{D}) = 1$.*

Let $\mathcal{S}(\mathcal{D}) = \{e_i : \phi_i(\mathcal{D}) = 1\}$ be the *solved set*—entities whose generated code passes all tests. The goal is to find documentation maximizing the number of solved entities:

$$\mathcal{D}^* = \arg\max_{\mathcal{D}} |\mathcal{S}(\mathcal{D})|. \qquad (2)$$

This formulation naturally yields a search problem: test execution provides a computable objective, and test failures offer rich diagnostic signals—error messages reveal type mismatches, assertion failures expose incorrect outputs, and stack traces indicate where execution diverged. These signals directly reveal the agent's *knowledge gap*: what it misunderstood that caused incorrect generation. The optimization task is to navigate documentation space, letting errors guide refinement toward descriptions that enable correct code synthesis.

## 2.2. The Whack-a-Mole Phenomenon

Code generation introduces a challenge absent in typical search problems: the outputs are *coupled* through the call graph.

**Definition 2** (Output Coupling). *A generation task exhibits output coupling if the correctness of $c_i$ depends not only on its specification $d_i$ but also on the actual realizations of outputs it depends on:*

$$\phi_i(\mathcal{D}) = \phi_i(d_1, \ldots, d_n, c_1, \ldots, c_n). \qquad (3)$$

When entity $e_i$ calls $e_j$, the correctness of $c_i$ depends on $c_j$'s *actual implementation*—not merely its documentation $d_j$. Refining $d_j$ to improve $c_j$ may produce a different (though correct) implementation, which breaks $c_i$ even when $d_i$ remains unchanged. This coupling propagates through the call graph, producing the **whack-a-mole phenomenon**: improving one entity breaks another; fixing that breaks a third; the search oscillates without convergence.

## 2.3. Limitations of Self-Refinement

The dominant paradigm for LLM-based code improvement is *self-refinement*: the model iteratively refines its outputs based on execution feedback (Madaan et al., 2023; Chen et al., 2024b). While effective for independent tasks, this paradigm exhibits two failure modes under output coupling.

Self-refinement selects entities to refine without reasoning over dependency structure—typically at random or by prioritizing entities with the most errors. This ignores the call graph: when entity $e_i$ calls $e_j$, refining $e_j$ may change its implementation, breaking $e_i$ even when $e_i$'s documentation is unchanged. We call this **refinement interference**.

**Definition 3** (Refinement Interference). *Let $\mathcal{D}[d_i \mapsto d_i']$ denote documentation with $d_i$ replaced by $d_i'$. A refinement $d_i \to d_i'$ exhibits* interference *if it degrades other entity:*

$$\exists j \neq i : \quad \phi_j(\mathcal{D}[d_i \mapsto d_i']) < \phi_j(\mathcal{D}) \qquad (4)$$

Output coupling makes interference pervasive rather than exceptional, producing characteristic oscillation—a whack-a-mole cycle where fixing one entity breaks another, preventing convergence.

**Theorem 1** (Suboptimality of Self-Refinement). *For any $n \geq 2$, there exists a problem instance where self-refinement achieves $|\mathcal{S}|/|\mathcal{S}^*| = O(1/n)$, where $\mathcal{S}^*$ denotes the optimal solved set.*

The proof constructs an adversarial star-graph instance; refer to Appendix §A.1 for details.

Even when interference is avoided through careful ordering, a second challenge remains: finding refinements that actually improve the target entity. We term this **homogeneous exploration**. Self-refinement conditions updates on all error messages simultaneously. Since test cases are LLM-generated, their quality varies: some are *informative* (revealing true knowledge gaps), some are *uninformative* (providing no learning signal), and some are *noisy* (misleading the refinement direction). When noisy errors co-occur, the LLM infers spurious patterns and produces overfitted documentation that addresses artifacts rather than genuine deficiencies. Moreover, repeated sampling under identical error context yields homogeneous candidates, trapping the search in local optima. We formalize this phenomenon through an ICL-theoretic analysis in §3.3, showing that refinement quality degrades predictably with error noise.

Escaping these limitations requires: (i) *dependency-guided traversal* that processes callees before callers to prevent interference and ensure monotonic progress (§3.2), and (ii) *diversity-augmented exploration* that conditions refinements on different error subsets to increase the probability of finding high-quality solutions (§3.3).

## 3. DocSearch

### 3.1. Overview

DOCSEARCH is a self-improving framework that iteratively refines documentation guided by test feedback. We construct pseudo test cases to guide the search: an LLM generates complete test functions which are then validated by execution on the reference implementation, retaining only verified-correct tests. This enables a closed loop—$\mathcal{D} \to \mathcal{C} \to \mathbf{e} \to \mathcal{D}'$—where documentation produces code, test failures diagnose knowledge gaps, and targeted refinements yield improved documentation.

The central challenge is output coupling: refining one entity's documentation may change its implementation, breaking callers even when their documentation is unchanged. Escaping this whack-a-mole trap requires addressing two failure modes of iterative refinement (§2.3). First, *refinement interference*—refining an entity without considering

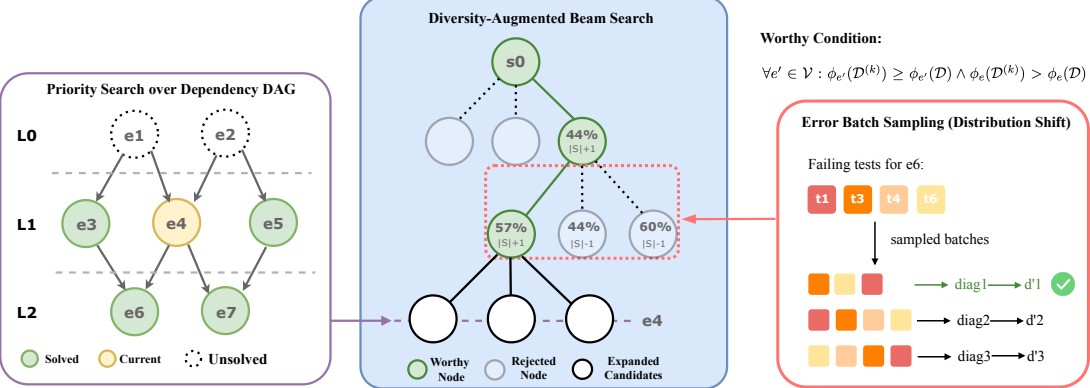

*Figure 2.* **Overview of DOCSEARCH.** The outer level (*left*) selects entities in reverse topological order via the call graph. The inner level (*center*) performs beam search over documentation refinements, committing only those satisfying the worthy condition (green path). Error batch sampling (*right*) induces diverse diagnoses by varying test subset composition, enabling escape from local optima.

dependencies may break its callers. We prevent this by processing entities in reverse topological order, ensuring that when refining an entity, all its callees have already stabilized. Second, *exploration collapse*—even when interference is avoided, conditioning on all error messages simultaneously causes refinements to concentrate around similar hypotheses. We escape this by conditioning each candidate on a different error batch, increasing the probability of sampling refinements that improve the target entity.

These mechanisms naturally decompose into a dependency-guided bi-level search (illustrated in Figure 2, refer to §B for details):

- **Outer level** (§3.2): A *priority search* over the dependency DAG determines *which entity* to refine, following reverse topological order with siblings prioritized by error count.
- **Inner level** (§3.3): A *diversity-augmented beam search* explores *how to refine* the selected entity, with each beam branch conditioned on a different error batch to escape local optima.
- **Worthy condition** (§3.4): A coupling mechanism governs *when to commit* refinements, ensuring monotonic progress by rejecting regressive updates.

## 3.2. Outer Level: Priority Search over Dependency DAG

The outer-level search determines which entity to refine next. The key constraint is output coupling: if entity $e_i$ calls $e_j$, then refining $e_j$ may change its implementation, potentially breaking $e_i$ even when $e_i$'s documentation is unchanged. We eliminate downstream interference by processing entities in reverse topological order—an entity becomes eligible for refinement only when all its callees have been finalized (solved or marked intractable). By the time we refine entity $e$, everything $e$ depends on has stabilized, so our refinement cannot propagate downward. Interference is thus confined to *siblings*: entities at the same topological level that share

common callees. Among eligible siblings, we prioritize those with fewer failing tests, as entities with fewer errors likely have simpler knowledge gaps, and solving them first provides stable context for harder siblings. This combination of reverse topological order and error-count priority defines a deterministic traversal that processes the dependency DAG systematically from leaves to roots.

## 3.3. Inner Level: Diversity-Augmented Beam Search

Given the selected entity $e$ and its failing tests $\mathcal{F}$, the inner-level search explores *how to refine* the documentation by generating $W$ candidates in parallel and committing the first that meets the worthy condition.

**The Signal Corruption Problem.** The diagnostic value of failing tests is inherently heterogeneous. Some tests directly reveal the knowledge gap causing incorrect generation, while others fail due to cascading effects from dependencies or provide redundant signals that dilute the true pattern. We formalize this heterogeneity: the truly informative errors form only a *subset* of all observed failures.

**Definition 4** (Error Partition)**.** *Let $\mathcal{F}$ denote failing tests for entity $e$. We partition $\mathcal{F}$ into:*

- *$\mathcal{F}^+$ (*informative*): errors that directly reveal the knowledge gap $g^*$*
- *$\mathcal{F}^-$ (*noisy*): errors arising from cascading effects, redundant signals, or misleading patterns*

*This partition is* unknown *a priori, but $\mathcal{F}^+$ alone suffices to identify $g^*$.*

**Theorem 2** (Signal Corruption)**.** *Let $q(\mathbf{e}) = P(\text{successful refinement} \mid \mathbf{e})$ denote the success probability when conditioning on error batch $\mathbf{e}$. Under Assumptions 1–5 (Appendix §A.2), for batches $\mathbf{e}^+ \subseteq \mathcal{F}^+$ and $\mathbf{e}^- \subseteq \mathcal{F}^-$:*

$$q(\mathbf{e}^-) \ll q(\mathbf{e}^+) \qquad (5)$$

*Moreover, mixing noisy errors with informative ones degrades performance:* $q(\mathbf{e}^+ \cup \mathbf{e}^-) < q(\mathbf{e}^+)$ *when* $|\mathbf{e}^-| > 0$.

The proof is provided in Appendix §A.2.

**Diversified Batch Sampling.** Since the informative subset $\mathcal{F}^+$ is unknown, we increase the probability of encountering an informative-dominated batch through diversified sampling. We formalize refinement as $d' \sim P_\theta(\cdot \mid d, \mathbf{e})$, where $d$ is current documentation, $\mathbf{e}$ is the error batch, and $\theta$ denotes LLM parameters.

**Definition 5** (Sampling Strategies). *Given budget $W$ and failing tests $\mathcal{F}$:*

- Uniform sampling*: condition on all errors $\mathcal{F}$ and draw $W$ i.i.d. samples.*
- Diversified sampling*: construct $W$ distinct batches $\{\mathbf{e}_1, \ldots, \mathbf{e}_W\}$ via subset selection with varying composition, and draw one sample per batch.*

**Coverage Advantage of Diversified Sampling.** By Theorem 2, conditioning on the informative subset $\mathcal{F}^+$ achieves higher success probability than conditioning on the full set $\mathcal{F}$: $q(\mathcal{F}^+) > q(\mathcal{F})$. Since $\mathcal{F}^+$ is unknown a priori, uniform sampling draws all $W$ candidates conditioned on the same full set $\mathcal{F}$, with success probability $1 - (1 - q(\mathcal{F}))^W$. Diversified sampling instead spreads the $W$ candidates across distinct batches, so as long as at least one batch with success probability higher than $q(\mathcal{F})$ is reachable with non-negligible probability, the chance of obtaining at least one successful refinement is improved. In other words, diversified sampling acts as a *hedging strategy*—it covers more of the batch space, increasing the probability of encountering an informative-dominated configuration without requiring knowledge of the partition $\mathcal{F}^+/\mathcal{F}^-$.

### 3.4. Worthy Condition: Coupling the Two Levels

The worthy condition couples the outer and inner levels by determining when to commit refinements, ensuring monotonic progress throughout the search. The search process constructs a tree where each node $v$ represents a documentation state $\mathcal{D}_v$ with associated pass rates $\{\phi_e(\mathcal{D}_v)\}_{e \in \mathcal{E}}$; the root contains the initial documentation, and children of a node represent alternative refinements from different error batches. A node $v$ with parent $u$ is *worthy* if the refinement causes no regression: $\phi_e(\mathcal{D}_v) \geq \phi_e(\mathcal{D}_u)$ for all $e \in \mathcal{E}$. A *worthy path* is a root-to-node path where every node is worthy (illustrated in Figure 2). The worthy condition commits a refinement if and only if the resulting node lies on a worthy path. This guarantees monotonicity: along any worthy path $v_0 \to v_1 \to \cdots \to v_T$, the solved set $\mathcal{S}_v = \{e : \phi_e(\mathcal{D}_v) = 1\}$ satisfies $\mathcal{S}_{v_0} \subseteq \mathcal{S}_{v_1} \subseteq \cdots \subseteq \mathcal{S}_{v_T}$, directly preventing the whack-a-mole phenomenon—once an entity is solved, it remains solved.

### 3.5. Handling Realistic Dependency Structures

While the bi-level search in §3.2–§3.4 assumes a clean dependency DAG, real codebases may contain cyclic dependencies or imprecise call graphs from dynamic dispatch and indirect calls; DOCSEARCH extends to these settings with minimal additions. For *cycles*, a preprocessing step contracts each strongly connected component (SCC) into a meta-entity refined jointly, restoring a DAG over meta-entities; the worthy condition naturally extends by committing an SCC refinement only if no entity regresses, inside or outside the SCC. For *imprecise call graphs*, the worthy condition itself serves as a fallback detector: if refining $e_j$ triggers regression on a previously-solved $e_i$ whose dependency was not captured statically, the implicit edge $(e_i, e_j)$ is added to $G$, the topological order is updated, and the existing search proceeds as normal.

## 4. Experiments

### 4.1. Experimental Setup

**Benchmark.** We evaluate on DEVEVAL+, a curated benchmark for module-level code generation comprising **20 Python modules** with **227 entities** validated by **631 test cases**. Modules are sourced from DevEval (Li et al., 2025) and additional GitHub repositories, selected for diversity across application domains (finance, biological analysis, signal processing, etc.). The benchmark exhibits significant structural complexity: average dependency graph depth of 4.2, maximum depth of 9, providing a challenging testbed for studying output coupling effects (full details in Appendix §G).

**Baselines.** We compare against three categories:

- *Static methods*: RepoAgent (Luo et al., 2024) and DocAgent (Yang et al., 2025) generate documentation without test feedback, representing human-oriented documentation paradigms.
- *Commercial agents*: Cursor (Truell et al., 2022) and Claude Code (Anthropic, 2024), state-of-the-art coding assistants evaluated under four configurations: (1) *base*: generate documentation with default settings; (2) *+Topo*: provide reverse topological order as context to guide generation sequences; (3) *+Test*: prompt the agent to generate test cases from documentation, then iteratively refine code until tests pass; (4) *+Topo+Test*: combine both strategies. Cursor supports Gemini-2.5-Flash and Claude-4.5-Sonnet but not GPT-4o; Claude Code only supports Claude models.
- *Iterative refinement*: refines documentation using test feedback, selecting the next entity either randomly (Iterative) or in reverse topological order (Iterative+Topo).

For all comparisons, final results are reported using ground-

*Table 1.* Main results on DEVEVAL+. $|\mathcal{S}|$: Solve Rate (%). $\bar{\phi}$: Pass Rate (%). Best in **bold**, second best underlined.

| Method | Gemini-2.5-Flash | | Claude-4.5-Sonnet | | GPT-4o | |
|---|---|---|---|---|---|---|
| | $|\mathcal{S}|$ | $\bar{\phi}$ | $|\mathcal{S}|$ | $\bar{\phi}$ | $|\mathcal{S}|$ | $\bar{\phi}$ |
| *Static Methods (No Optimization)* | | | | | | |
| RepoAgent | 21.1 | 32.8 | 37.0 | 49.6 | 29.5 | 42.0 |
| DocAgent | 15.9 | 29.3 | 30.4 | 46.9 | 23.3 | 39.1 |
| *Commercial Coding Agents* | | | | | | |
| Cursor | 17.2 | 26.9 | 32.2 | 42.0 | – | – |
| + Topo | 22.9 | 32.8 | 38.8 | 48.2 | – | – |
| + Test | 33.9 | 42.0 | 50.2 | 54.7 | – | – |
| + Topo + Test | 42.7 | 49.8 | 55.1 | 59.4 | – | – |
| Claude Code | – | – | 41.9 | 52.3 | – | – |
| + Topo | – | – | 47.1 | 57.5 | – | – |
| + Test | – | – | 58.1 | 65.0 | – | – |
| + Topo + Test | – | – | 64.8 | 71.5 | – | – |
| *Iterative Refinement* | | | | | | |
| Iterative | 41.4 | 46.4 | 62.6 | 63.4 | 52.9 | 56.6 |
| + Topo | 44.1 | 48.7 | 67.4 | 67.5 | 58.1 | 60.9 |
| *Ours* | | | | | | |
| DOCSEARCH | **74.4** | **79.4** | **95.2** | **96.2** | **90.7** | **93.3** |

truth test cases, which are not visible to the search algorithms. Instead, the search is guided by LLM-generated tests. Details of test case generation are provided in Appendix §H; Implementation details and prompts for all baselines are provided in Appendix §F.

**Metrics.** We report **Solve Rate** $|\mathcal{S}|/n$ (proportion of entities achieving $\phi_i = 1$) as the primary metric, and **Pass Rate** $\bar{\phi} = \frac{1}{n}\sum_i \phi_i$ (average test pass rate) as a secondary metric capturing partial progress.

### 4.2. Main Results

Table 1 presents comprehensive comparisons across methods and base LLMs. DOCSEARCH achieves the highest solve rates across all three LLMs: 74.4% with Gemini-2.5-Flash, 95.2% with Claude-4.5-Sonnet, and 90.7% with GPT-4o. The consistency of improvements (27.8–32.6% absolute gains over the best iterative baseline) suggests that our approach addresses fundamental optimization challenges rather than exploiting model-specific behaviors. Below, we analyze why each baseline category falls short.

**Static Methods: The Documentation-Agent Gap.** RepoAgent and DocAgent achieve solve rates of only 15.9–37.0%, revealing a fundamental *documentation-agent gap*: documentation optimized for human readability often misleads LLMs into incorrect implementations. Human developers naturally fill in gaps through world knowledge and contextual reasoning—they understand that "returns a list of items" likely means maintaining input order, or that "processes the data" implies specific error handling conventions. By contrast, LLMs interpret documentation literally but resolve

ambiguities unpredictably, yielding different implementations from the same underspecified phrase. This gap cannot be closed by prompting alone and requires documentation optimized for agent comprehension.

**Commercial Agents: Missing Structural Mechanisms.** Cursor and Claude Code, even with test feedback and topological ordering (+Topo+Test), achieve only 55.1% and 64.8% respectively—substantially below DOCSEARCH. These tools incorporate sophisticated code understanding and generation capabilities, yet lack two mechanisms our analysis identifies as essential. First, without the *worthy condition*, they commit refinements that solve the current entity but inadvertently break others, accumulating regressions over iterations. Second, without *tree search*, they cannot backtrack from suboptimal early refinements, allowing a single poor decision to propagate through later iterations. This performance gap shows that raw model capacity is insufficient; principled search structure is required.

**Iterative Refinement: Ordering Alone Is Insufficient.** Iterative methods with test feedback achieve 41.4–67.4% solve rate—substantial improvement over static methods, confirming that execution feedback provides valuable signal. Adding topological ordering (+Topo) yields consistent but modest improvements (2.7–5.2%), validating that processing order matters for managing dependencies. However, even with correct ordering, iterative methods trail DOCSEARCH by 27.8–30.3%. This gap reveals the critical missing mechanism: the worthy condition. Without it, these methods commit refinements that solve the target entity but break siblings at the same topological level—entities that share common callees. The result is the characteristic oscillation predicted by Theorem 1: each iteration's gains are partially offset by regressions, preventing convergence to high solve rates.

**Cross-Model Consistency.** The improvement pattern is remarkably consistent across LLMs of varying capabilities. Despite substantial differences in base performance—Gemini-2.5-Flash achieves only 44.1% with the best iterative baseline while Claude-4.5-Sonnet reaches 67.4%. DOCSEARCH delivers comparable absolute gains across all models (+30.3%, +27.8%, and +32.6% respectively over Iterative+Topo). This consistency suggests that DOCSEARCH addresses a fundamental challenge in the optimization landscape—output coupling—rather than compensating for specific model weaknesses. Stronger models achieve higher absolute performance but face the same structural barriers that DOCSEARCH overcomes.

### 4.3. Escaping the Whack-a-Mole Trap

Having established DOCSEARCH's effectiveness, we now analyze *how* it escapes the whack-a-mole trap that limits iterative approaches.

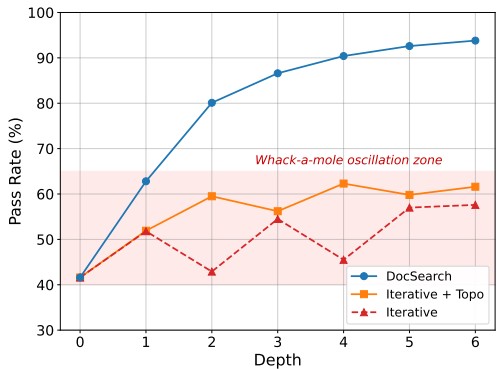

*Figure 3.* Pass rate trajectories on a 3-module subset of DEVEVAL+ (GPT-4o). The $x$-axis denotes the number of entities processed; the $y$-axis denotes cumulative pass rate. Iterative refinement (orange) oscillates due to regressions, while DOCSEARCH (blue) achieves monotonic progress.

*Table 2.* Ablation study (GPT-4o) on a subset of DEVEVAL+ with 10 modules. Each row removes one component from the full DOCSEARCH.

| Configuration | $|\mathcal{S}|$ (%) | $\bar{\phi}$ (%) | $\Delta|\mathcal{S}|$ |
|---|---|---|---|
| DOCSEARCH (full) | **90.7** | **93.3** | – |
| w/o Reverse Topo | 72.2 | 75.8 | -18.5 |
| w/o Error Diversity | 81.1 | 83.5 | -9.6 |
| w/o Worthy Condition | 83.7 | 85.9 | -7.0 |

### 4.3.1. VISUALIZING THE PHENOMENON

Figure 3 directly visualizes the whack-a-mole phenomenon on a representative subset of 3 modules (34 entities). Both iterative variants exhibit characteristic oscillation within the shaded "whack-a-mole zone": Iterative+Topo (orange) fluctuates between 52–62%, while Iterative without topological ordering (red, dashed) shows even more severe oscillation between 43–58%, with gains offset by regressions—this reflects systematic interference from output coupling (Definition 2), not random noise. When a callee's documentation is refined, its implementation may change, breaking callers that implicitly depended on specific details. In contrast, DOCSEARCH (blue) achieves monotonic improvement: the worthy condition ensures each committed refinement maintains or increases the solved set, breaking the oscillation cycle and converging to over 90% pass rate.

### 4.3.2. COMPONENT CONTRIBUTION ANALYSIS

Table 2 isolates each component's contribution. The largest degradation ($-18.5\%$) comes from removing **reverse topological order**, confirming it as the structural foundation: it prevents downstream interference by ensuring callees are finalized before callers, confining interference to siblings rather than the entire call chain. **Error diversity** ($-9.6\%$) addresses exploration collapse: when all candidates condition on identical error batches,

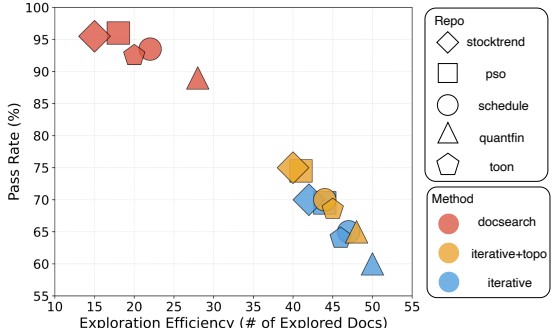

*Figure 4.* Exploration efficiency: pass rate vs. number of explored documentation variants. DOCSEARCH (red) clusters in the top-left (high performance, few explorations), while Iterative+Topo (orange) and Iterative (blue) require more exploration for inferior results. Each point represents one module.

they induce similar hypotheses and fail together if signals are misleading; diversified sampling increases the probability that at least one batch reveals the true pattern. The **worthy condition**'s smaller degradation ($-7.0\%$) reflects that reverse topological order already eliminates most interference—the remaining impact comes from sibling conflicts (entities at the same level sharing common callees), which the worthy condition resolves by rejecting refinements that solve one entity but break its siblings.

### 4.3.3. EXPLORATION EFFICIENCY

Figure 4 plots pass rate against the number of explored documentation variants, revealing a striking efficiency gap. DOC-SEARCH points cluster in the *top-left corner*—achieving 88–97% pass rate while exploring only 15–25 variants per module. Iterative+Topo, despite exploring 1.5–2× more variants, achieves inferior performance that plateaus around 70–75%. Iterative without topological ordering performs worst, requiring 40–50 explorations while achieving only 59–70% pass rate.

This efficiency gap has two sources. First, *error-driven refinement* targets specific knowledge gaps rather than exploring randomly. Each refinement addresses a diagnosed deficiency, making exploration purposeful rather than diffuse. Second, *reverse topological ordering* eliminates wasted effort on refinements that would be invalidated by subsequent downstream changes. In iterative approaches, refining entity $e_i$ can be undone if a later refinement of callee $e_j$ reverses it, wasting the effort on $e_i$. A complementary token-efficiency analysis is provided in §4.6.

### 4.4. Downstream Utility: Cross-Language Generation

Beyond improving same-language code generation, can the synthesized playbook serve broader purposes? We explore whether it functions as a language-agnostic playbook — a behavioral specification that, although optimized for one

*Table 3.* Cross-language generation with GPT-4o. Python→Java is evaluated on the 10-module subset of DEVEVAL+; Python→Go on a 5-module subset. Doc-only substantially outperforms Source-only across both targets, suggesting the synthesized documentation serves as an effective language-agnostic playbook.

| Input Configuration | Python→Java | | Python→Go | |
|---|---|---|---|---|
| | $\|\mathcal{S}\|$ (%) | $\bar{\phi}$ (%) | $\|\mathcal{S}\|$ (%) | $\bar{\phi}$ (%) |
| Source-only | 41.4 | 52.3 | 39.6 | 44.7 |
| Doc-only (optimized) | 68.7 | 74.2 | 62.3 | 68.9 |
| Doc+Source | **79.3** | **83.6** | **72.1** | **78.4** |

source language, enables code generation in languages other than the source.

**Experimental Setup.** We evaluate cross-language transfer from Python to two target languages, Java and Go, on subsets of our suite (10 modules for Python→Java and 5 modules for Python→Go). We select modules whose semantics are amenable to cross-language transfer (see Appendix G.4 for details). Documentation is first optimized by DOCSEARCH for Python code generation, and test harnesses are then manually rewritten in the target language while preserving original input–output pairs to ensure semantic equivalence. We compare three configurations: **Source-only** (direct cross-language translation from Python source), **Doc-only** (generation from optimized documentation), and **Doc+Source** (using both documentation and source code).

**Results and Analysis.** Table 3 reveals a consistent pattern across both target languages: Doc-only substantially outperforms Source-only (Java: $68.7\%$ vs. $41.4\%$, $+27.3$; Go: $62.3\%$ vs. $39.6\%$, $+22.7$). This gap reflects a fundamental difference in representations: source code describes *how* to compute using Python-specific idioms (list comprehensions, duck typing), requiring the LLM to simultaneously understand the idioms, identify intent, and map to target-language equivalents; documentation describes *what* the function computes, abstracting away Python-specific details and freeing the LLM to choose target-appropriate implementations. The Doc+Source combination achieves the best results on both targets ($79.3\%$ and $72.1\%$), as documentation specifies behavioral intent while source code conveys precise computations difficult to describe in natural language—together providing both the "what" and the "how". The fact that Java and Go—two languages with different paradigms (object-oriented vs. structural, exception-based vs. error-value)—exhibit the same ordering and comparable improvement magnitudes suggests that DOCSEARCH-optimized documentation captures language-agnostic behavioral specifications rather than patterns specific to any single target.

### 4.5. Human Evaluation of Documentation Quality

While DOCSEARCH optimizes documentation against downstream code correctness rather than human readability, we conduct a paired human evaluation to test whether the opti-

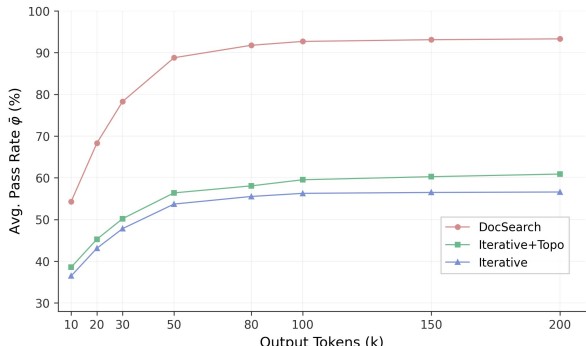

*Figure 5.* Token efficiency on DEVEVAL+ (GPT-4o): cross-module average pass rate $\bar{\phi}$ as a function of cumulative output tokens.

mized documentation is also more useful to human readers.

**Protocol.** We randomly select 30 entities from 6 modules in DEVEVAL+ where the documentation changed substantively between the initial and DOCSEARCH-optimized versions (GPT-4o, $B=50$). Five annotators with computer-science backgrounds independently rate each documentation pair, blinded to version identity with presentation order randomized across entities. Each documentation is rated on five 1–5 Likert dimensions—*Completeness*, *Correctness*, *Clarity*, *Helpfulness*, and *Specificity* (full rubric in Appendix C)—and we report Krippendorff's ordinal $\alpha$ as the inter-annotator agreement measure.

*Table 4.* Paired human evaluation of documentation quality. Scores are mean $\pm$ standard deviation over 30 entities, each rated by 5 annotators. $\alpha$: Krippendorff's ordinal $\alpha$.

| Dimension | Initial | Optimized | $\alpha$ |
|---|---|---|---|
| Completeness | $3.10 \pm 1.03$ | $\mathbf{4.67 \pm 0.54}$ | 0.94 |
| Correctness | $3.58 \pm 0.86$ | $\mathbf{4.50 \pm 0.85}$ | 0.78 |
| Clarity | $2.88 \pm 0.91$ | $\mathbf{4.42 \pm 0.67}$ | 0.92 |
| Helpfulness | $2.40 \pm 1.16$ | $\mathbf{4.43 \pm 0.80}$ | 0.71 |
| Specificity | $2.17 \pm 1.05$ | $\mathbf{4.52 \pm 0.85}$ | 0.75 |

**Results and Analysis.** Table 4 reports per-dimension scores. Optimized documentation improves across all five dimensions, with the largest gains in *Specificity* ($+2.35$) and *Helpfulness* ($+2.03$)—the two dimensions most tied to actionable behavioral information—followed by *Completeness* ($+1.57$) and *Clarity* ($+1.54$). Inter-annotator agreement is high ($\alpha \geq 0.71$), indicating these improvements are not artifacts of annotator subjectivity. The pattern aligns with the qualitative shift observed throughout optimization: DOCSEARCH moves documentation from high-level descriptions of *what* an entity does to precise specifications of *how* it behaves (example in Appendix D); the gain in *Correctness* ($+0.92$) further suggests that test-driven refinement also corrects subtle inaccuracies.

### 4.6. Cost Worthiness

We acknowledge that DOCSEARCH's beam search and worthy condition involve multiple generation cycles. However, a token-level analysis reveals that this apparent overhead actually yields superior efficiency. Figure 5 tracks $\bar{\phi}$ as a function of cumulative output tokens: at just 20k tokens, DOCSEARCH (68.3%) already exceeds Iterative's final performance at 200k tokens (56.6%), a $10\times$ efficiency gap. Moreover, DOCSEARCH is a one-time offline optimization: the optimized documentation is reused across all downstream tasks without re-incurring the search cost.

## 5. Related Work

### 5.1. Code Documentation

Code documentation generation has evolved through three paradigms. Early *template-based* approaches relied on rule-driven heuristics and predefined templates to extract structural information from source code (Haiduc et al., 2010; Sridhara et al., 2010; Moreno et al., 2013). *Training-based* methods subsequently emerged, leveraging neural architectures such as LSTMs with attention (Iyer et al., 2016) and Transformer-based models (Ahmad et al., 2020) trained on code-comment pairs for function-level summarization. More recently, *LLM-based* approaches have scaled to repository-level documentation: RepoAgent (Luo et al., 2024) leverages global structure analysis and reference relationships, while DocAgent (Yang et al., 2025) employs multi-agent collaboration with topological code processing. However, these methods remain fundamentally *human-oriented*—quality is assessed through subjective metrics like completeness and helpfulness (Luo et al., 2024; Yang et al., 2025), or text similarity measures such as BLEU and ROUGE (Roy et al., 2021). As LLMs increasingly serve as autonomous coding agents, documentation must be optimized for *agent comprehension*, not just human readability (Hong et al., 2024; Qian et al., 2024). To our knowledge, DOCSEARCH is the first to formulate *agent-oriented documentation generation*, measuring quality by downstream code correctness rather than subjective human assessment.

### 5.2. LLM-Based Test Generation

LLM-based test generation has emerged as a practical alternative when ground-truth tests are unavailable. Early work shows that LLMs can produce coverage-driven test suites (Schäfer et al., 2024; Chen et al., 2024c), though generated tests often cluster around similar patterns and miss critical edge cases (Jain et al., 2025). Coverage-guided sampling (Lemieux et al., 2023) and interactive test-driven refinement (Fakhoury et al., 2024) have been proposed to improve diversity and accuracy. Beyond standalone test generation, recent work uses LLM-generated tests as feedback for code synthesis (Chen et al., 2023; Zhang et al., 2023b) and as the verification signal for reconstructing entire repositories: ProgramBench (Yang et al., 2026) generates behavioral tests via agent-driven fuzzing to evaluate whole-program rebuilds, and RepoZero (Zhang et al., 2026) performs iterative LLM-based test generation as part of its agentic code-test evolution loop to verify repository-level outputs. DOCSEARCH validates LLM-generated tests against the reference implementation, preserving both oracle correctness and diagnostic signal.

### 5.3. Inference Time Scaling

Inference-time scaling has become a central paradigm for improving LLM reasoning at test time. Early work focused on prompting strategies that elicit intermediate reasoning, such as chain-of-thought and tree-based prompting (Wei et al., 2022; Yao et al., 2023). Beyond prompting, recent efforts explore sampling-based search and reward-guided aggregation to further boost inference performance (Lightman et al., 2024; Brown et al., 2024; Snell et al., 2025; Wu et al., 2025; Light et al., 2025c;b; Liu et al., 2025b), including REBASE (Wu et al., 2025), which uses a process reward model (PRM) to prioritize promising reasoning trajectories. More recently, multi-agent frameworks scale inference via coordination and iterative self-improvement, exemplified by SWE-Search (Antoniades et al., 2025), which combines MCTS with multi-agent learning (Jin et al., 2025; Light et al., 2025a; Zhang et al., 2025b;c;a). However, these approaches cannot be directly applied to agent-oriented documentation generation due to the output coupling problem.

## 6. Conclusion

In this paper, we introduced DOCSEARCH, a dependency-guided bi-level search framework for synthesizing repo-specific playbooks for coding agents, framing playbook optimization as a black-box optimization problem driven by downstream code correctness. By explicitly modeling call dependencies through a priority search and exploiting test-time feedback via a beam tree search with diversified error-message sampling, DOCSEARCH avoids the output-coupling–induced whack-a-mole phenomenon that hinders conventional refinement methods. We provided theoretical guarantees of monotonic progress under a worthy condition and showed how diversified diagnostic signals help escape local optima during inference-time optimization. Empirically, DOCSEARCH achieves substantial improvements on DEVEVAL+, consistently outperforming strong baselines across multiple LLMs, exhibiting stable convergence behavior, and enabling effective cross-language transfer. These results suggest that the synthesized playbooks serve as language-agnostic, repo-specific specifications for agentic code generation.

## Acknowledgements

This work is supported in part by the National Science Foundation under grant 2442171 and the Google Academic Research Award (GARA). Any opinions, findings, and conclusions made in this paper are those of the authors and do not necessarily reflect the views of the funding agencies.

## Impact Statement

This paper proposes a method for improving the reliability and effectiveness of large language models as autonomous coding agents by optimizing code documentation through inference-time search. By enabling more systematic and stable self-improvement without additional human supervision, this work has the potential to reduce development costs, improve software quality, and support broader adoption of LLM-based programming tools. At the same time, more capable coding agents may lower the barrier to producing software at scale, which could amplify both beneficial and harmful applications. However, our approach does not introduce new capabilities beyond code document generation and relies on existing test-based evaluation, and we do not anticipate novel ethical risks beyond those already associated with LLM-based software development. Overall, this work aims to advance the field of machine learning, and we believe its societal implications are aligned with well-established considerations in deploying automated coding systems.

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

# A. Proofs

This appendix provides complete proofs for the theoretical results stated in the main paper.

## A.1. Proof of Theorem 1 (Greedy Suboptimality)

**Theorem** (Greedy Suboptimality). *For any $n \geq 2$, there exists a problem instance where:*

$$\frac{|\mathcal{S}^{\text{GREEDY}}|}{|\mathcal{S}^*|} = O\left(\frac{1}{n}\right) \tag{6}$$

*Proof.* We construct an adversarial instance using a star graph topology.

**Construction.** Consider $n$ entities $\mathcal{E} = \{e_1, e_2, \ldots, e_n\}$ with call graph $G$ where $e_1$ calls all other entities: $(e_1, e_j) \in E$ for $j = 2, \ldots, n$. Each entity has a binary documentation space $\mathcal{D}_i = \{d_i^0, d_i^1\}$.

Define the pass rate functions as:

$$\phi_1(\mathcal{D}) = \begin{cases} 1 & \text{if } d_1 = d_1^1 \text{ and } \forall j > 1 : d_j = d_j^0 \\ 0 & \text{otherwise} \end{cases} \tag{7}$$

$$\phi_j(\mathcal{D}) = \begin{cases} 0 & \text{if } d_1 = d_1^1 \\ 1 & \text{if } d_1 = d_1^0 \text{ and } d_j = d_j^1 \quad \text{for } j > 1 \\ 0 & \text{otherwise} \end{cases} \tag{8}$$

**Key Property.** The construction encodes a fundamental conflict: $e_1$ can only be solved when all leaves use $d_j^0$, but each leaf $e_j$ can only be solved when it uses $d_j^1$ and $e_1$ uses $d_1^0$. Crucially, once $d_1 = d_1^1$ is committed, all leaves become permanently unsolvable regardless of their documentation.

**Greedy Execution.** Starting from the initial state $s_0 = (d_1^0, d_2^0, \ldots, d_n^0)$, all pass rates are zero by construction, so $|\mathcal{S}(s_0)| = 0$. Greedy evaluates each single-step modification: flipping $d_1^0 \to d_1^1$ achieves $\phi_1 = 1$ (gain of one entity), and flipping any $d_j^0 \to d_j^1$ for $j > 1$ likewise achieves $\phi_j = 1$ (gain of one entity). All options yield the same immediate gain; suppose tie-breaking selects $e_1$. Greedy then commits to $s_1 = (d_1^1, d_2^0, \ldots, d_n^0)$ with $|\mathcal{S}(s_1)| = 1$.

At $s_1$, entity $e_1$ is solved ($\phi_1(s_1) = 1$), while $\phi_j(s_1) = 0$ for every $j > 1$. Crucially, the construction makes this state absorbing for the leaves: as long as $d_1 = d_1^1$, we have $\phi_j = 0$ regardless of $d_j$, so no single-step modification can increase the solved set. Greedy therefore terminates with $|\mathcal{S}^{\text{GREEDY}}| = 1$.

**Optimal Solution.** Consider the alternative state $s^* = (d_1^0, d_2^1, \ldots, d_n^1)$. Here $\phi_1(s^*) = 0$, since the condition for solving $e_1$ requires all leaves to use $d_j^0$, which is violated. Every leaf, however, is solved: $\phi_j(s^*) = 1$ for all $j > 1$, giving $|\mathcal{S}^*| = n - 1$.

**Approximation Ratio.**

$$\frac{|\mathcal{S}^{\text{GREEDY}}|}{|\mathcal{S}^*|} = \frac{1}{n-1} = O\left(\frac{1}{n}\right) \tag{9}$$

This ratio approaches zero as $n \to \infty$, completing the proof.

**Remark 1** (Robustness of the Construction). *The construction can be extended to handle randomized tie-breaking by making $e_1$'s improvement slightly larger (e.g., $\phi_1 = 1$ vs $\phi_j = 1 - \epsilon$ for arbitrarily small $\epsilon$). The approximation ratio remains $O(1/n)$.*

## A.2. Proof of Theorem 2

We restate Theorem 2 for completeness.

**Theorem** (Signal Corruption, restatement of Theorem 2). *Let $\mathcal{F}$ be the set of failing tests for an entity, partitioned into $\mathcal{F} = \mathcal{F}^+ \cup \mathcal{F}^-$, where $\mathcal{F}^+$ are informative errors and $\mathcal{F}^-$ are noisy errors (Definition 4). Let $q(\mathbf{e}) = P(\text{successful refinement} \mid \mathbf{e})$ denote the success probability when conditioning on an error batch $\mathbf{e}$. Under Assumptions 1–5 below, for any batches $\mathbf{e}^+ \subseteq \mathcal{F}^+$ and $\mathbf{e}^- \subseteq \mathcal{F}^-$, we have*

$$q(\mathbf{e}^-) \ll q(\mathbf{e}^+), \tag{10}$$

*and mixing noisy errors with informative ones strictly degrades performance:*

$$q(\mathbf{e}^+ \cup \mathbf{e}^-) < q(\mathbf{e}^+) \qquad \text{whenever } |\mathbf{e}^-| > 0. \tag{11}$$

**Setup.** We model refinement as diagnosing a latent knowledge gap. Let $\mathcal{G}$ be a finite set of candidate knowledge gaps, with the ground-truth gap denoted by $g^* \in \mathcal{G}$. Each error message $e$ is treated as an observation generated from a likelihood $P(e \mid g)$.

**Assumption 1** (Conditional Independence within a Batch). *Conditioned on $g \in \mathcal{G}$, errors in a batch are independent: for any batch $\mathbf{e} = \{e_1, \ldots, e_m\}$,*

$$P(\mathbf{e} \mid g) = \prod_{t=1}^{m} P(e_t \mid g). \tag{12}$$

**Assumption 2** (Informative Separation). *There exists a constant $\gamma \in (0, 1)$ such that for every incorrect gap $g \neq g^*$ and every informative error $e \in \mathcal{F}^+$,*

$$\frac{P(e \mid g)}{P(e \mid g^*)} \leq \gamma. \tag{13}$$

*Equivalently, each informative error provides at least $-\log \gamma > 0$ log-likelihood advantage to the true gap.*

**Assumption 3** (Noise Non-Discriminativity (Adversarial-to-True)). *For every incorrect gap $g \neq g^*$ and every noisy error $e \in \mathcal{F}^-$,*

$$\frac{P(e \mid g)}{P(e \mid g^*)} \geq 1. \tag{14}$$

*That is, noisy errors provide no evidence in favor of $g^*$ and can weakly favor incorrect gaps.*

**Assumption 4** (Posterior-based Diagnosis). *The diagnosis step selects $\hat{g}(\mathbf{e}) = \arg\max_{g \in \mathcal{G}} P(g \mid \mathbf{e})$ under a prior $\pi(g)$ that is strictly positive on all $g \in \mathcal{G}$.*

**Assumption 5** (Success Monotone in Correct Identification). *There exist constants $0 < a \leq b \leq 1$ such that for any batch $\mathbf{e}$,*

$$a \cdot P(\hat{g}(\mathbf{e}) = g^* \mid \mathbf{e}) \ \leq \ q(\mathbf{e}) \ \leq \ b \cdot P(\hat{g}(\mathbf{e}) = g^* \mid \mathbf{e}). \tag{15}$$

*This captures that correctly identifying the knowledge gap is necessary and approximately sufficient for producing a successful refinement.*

**Key Lemma: posterior odds bound.**

**Lemma 1** (Posterior Odds under Informative vs. Noisy Errors). *Under Assumptions 1–4, for any $g \neq g^*$ and any batch $\mathbf{e}$,*

$$\frac{P(g \mid \mathbf{e})}{P(g^* \mid \mathbf{e})} = \frac{\pi(g)}{\pi(g^*)} \cdot \prod_{e \in \mathbf{e}} \frac{P(e \mid g)}{P(e \mid g^*)}. \tag{16}$$

*In particular, for $\mathbf{e}^+ \subseteq \mathcal{F}^+$ and $\mathbf{e}^- \subseteq \mathcal{F}^-$,*

$$\frac{P(g \mid \mathbf{e}^+)}{P(g^* \mid \mathbf{e}^+)} \leq \frac{\pi(g)}{\pi(g^*)} \cdot \gamma^{|\mathbf{e}^+|}, \qquad \frac{P(g \mid \mathbf{e}^+ \cup \mathbf{e}^-)}{P(g^* \mid \mathbf{e}^+ \cup \mathbf{e}^-)} \geq \frac{P(g \mid \mathbf{e}^+)}{P(g^* \mid \mathbf{e}^+)}. \tag{17}$$

*Proof.* The posterior-odds identity follows from Bayes rule:

$$P(g \mid \mathbf{e}) = \frac{\pi(g) P(\mathbf{e} \mid g)}{\sum_{h \in \mathcal{G}} \pi(h) P(\mathbf{e} \mid h)},$$

hence

$$\frac{P(g \mid \mathbf{e})}{P(g^* \mid \mathbf{e})} = \frac{\pi(g)P(\mathbf{e} \mid g)}{\pi(g^*)P(\mathbf{e} \mid g^*)}.$$

Assumption 1 gives $P(\mathbf{e} \mid g) = \prod_{e \in \mathbf{e}} P(e \mid g)$, yielding the product form.

For the informative bound, apply Assumption 2 to each $e \in \mathbf{e}^+$: $\prod_{e \in \mathbf{e}^+} \frac{P(e|g)}{P(e|g^*)} \leq \gamma^{|\mathbf{e}^+|}$.

For the monotonicity under adding noisy errors, write

$$\frac{P(g \mid \mathbf{e}^+ \cup \mathbf{e}^-)}{P(g^* \mid \mathbf{e}^+ \cup \mathbf{e}^-)} = \frac{\pi(g)}{\pi(g^*)}\Big( \prod_{e \in \mathbf{e}^+} \frac{P(e \mid g)}{P(e \mid g^*)} \Big)\Big( \prod_{e \in \mathbf{e}^-} \frac{P(e \mid g)}{P(e \mid g^*)} \Big).$$

Assumption 3 implies each factor in the last product is $\geq 1$, so the whole ratio is $\geq$ the ratio without $\mathbf{e}^-$.

*Proof of Theorem 2.* We prove two claims: (i) noisy-only batches have low identification probability compared to informative batches; (ii) adding noisy errors degrades identification, hence degrades $q(\cdot)$.

**Step 1: Informative errors drive posterior mass to $g^*$.**   Fix any $g \neq g^*$. By Lemma 1,

$$\frac{P(g \mid \mathbf{e}^+)}{P(g^* \mid \mathbf{e}^+)} \leq \frac{\pi(g)}{\pi(g^*)}\gamma^{|\mathbf{e}^+|}.$$

Summing over all $g \neq g^*$ and using $\sum_{g \neq g^*} P(g \mid \mathbf{e}^+) = 1 - P(g^* \mid \mathbf{e}^+)$, we obtain

$$1 - P(g^* \mid \mathbf{e}^+) = \sum_{g \neq g^*} P(g \mid \mathbf{e}^+) \tag{18}$$

$$\leq P(g^* \mid \mathbf{e}^+) \sum_{g \neq g^*} \frac{\pi(g)}{\pi(g^*)}\gamma^{|\mathbf{e}^+|}. \tag{19}$$

Rearranging yields

$$P(g^* \mid \mathbf{e}^+) \geq \frac{1}{1 + \Big( \sum_{g \neq g^*} \frac{\pi(g)}{\pi(g^*)} \Big)\gamma^{|\mathbf{e}^+|}}, \tag{20}$$

which increases monotonically with $|\mathbf{e}^+|$ and approaches 1 as $|\mathbf{e}^+| \to \infty$.

**Step 2: Noisy errors cannot increase identification of $g^*$ and can decrease it.**   Consider $\mathbf{e}^+ \subseteq \mathcal{F}^+$ and any non-empty $\mathbf{e}^- \subseteq \mathcal{F}^-$ with $|\mathbf{e}^-| > 0$. By Lemma 1, for every $g \neq g^*$,

$$\frac{P(g \mid \mathbf{e}^+ \cup \mathbf{e}^-)}{P(g^* \mid \mathbf{e}^+ \cup \mathbf{e}^-)} \geq \frac{P(g \mid \mathbf{e}^+)}{P(g^* \mid \mathbf{e}^+)}.$$

Equivalently, the posterior odds against $g^*$ weakly increase for all incorrect gaps when adding $\mathbf{e}^-$. This implies the posterior mass on $g^*$ cannot increase:

$$P(g^* \mid \mathbf{e}^+ \cup \mathbf{e}^-) \leq P(g^* \mid \mathbf{e}^+). \tag{21}$$

Moreover, if there exists at least one incorrect $g$ and at least one noisy error $e \in \mathbf{e}^-$ such that $\frac{P(e|g)}{P(e|g^*)} > 1$, then the inequality is strict:

$$P(g^* \mid \mathbf{e}^+ \cup \mathbf{e}^-) < P(g^* \mid \mathbf{e}^+). \tag{22}$$

**Step 3: From posterior identification to refinement success.**   By Assumption 4, the probability of correct identification $P(\hat{g}(\mathbf{e}) = g^* \mid \mathbf{e})$ is monotone in $P(g^* \mid \mathbf{e})$ (ties can be broken deterministically). Therefore Step 2 implies

$$P(\hat{g}(\mathbf{e}^+ \cup \mathbf{e}^-) = g^* \mid \mathbf{e}^+ \cup \mathbf{e}^-)P(\hat{g}(\mathbf{e}^+) = g^* \mid \mathbf{e}^+), \tag{23}$$

whenever $|\mathbf{e}^-| > 0$ and some noisy error strictly favors an incorrect gap. Applying Assumption 5 yields

$$q(\mathbf{e}^+ \cup \mathbf{e}^-) < q(\mathbf{e}^+). \tag{24}$$

**Step 4: Comparing noisy-only vs. informative batches.** A noisy-only batch $\mathbf{e}^- \subseteq \mathcal{F}^-$ provides no advantage to $g^*$ by Assumption 3, hence it cannot drive $P(g^* \mid \mathbf{e}^-)$ close to 1. In contrast, Step 1 shows that with sufficiently many informative errors, $P(g^* \mid \mathbf{e}^+)$ can be made arbitrarily close to 1. By Assumption 5, the same separation translates to success probability, establishing $q(\mathbf{e}^-) \ll q(\mathbf{e}^+)$. This completes the proof.

**Remark 2.** *If a weaker noise model is preferred, Assumption 3 can be relaxed to $\frac{P(e|g)}{P(e|g^*)} \geq \exp(-\eta)$ for some $\eta \geq 0$ (approximately non-discriminative). Then Step 2 yields a quantitative bound: $P(g^* \mid \mathbf{e}^+ \cup \mathbf{e}^-) \leq \frac{P(g^*|\mathbf{e}^+)}{P(g^*|\mathbf{e}^+)+(1-P(g^*|\mathbf{e}^+))\exp(-\eta|\mathbf{e}^-|)}$, and the degradation remains monotone in $|\mathbf{e}^-|$.*

# B. Algorithm Details

Algorithm 1 presents the complete DOCSEARCH procedure with all subroutines.

---

**Algorithm 1** DOCSEARCH: Complete Bi-level Search

---

**Require:** Initial documentation $\mathcal{D}^{(0)}$, call graph $G$, test suite $\mathcal{T}$, width $W$, budget $B$
**Ensure:** Optimized documentation $\mathcal{D}^*$
1: **// Initialization**
2: $\mathcal{C}^{(0)} \leftarrow$ GENERATE$(\mathcal{D}^{(0)})$              ▷ Generate initial code
3: $\phi^{(0)} \leftarrow$ EVALUATE$(\mathcal{C}^{(0)}, \mathcal{T})$              ▷ Run all tests
4: $\mathcal{S}^{(0)} \leftarrow \{e_i : \phi_i^{(0)} = 1\}$            ▷ Initially solved entities
5: $\mathcal{I}^{(0)} \leftarrow \emptyset$               ▷ Initially no intractable entities
6: $v_{\text{root}} \leftarrow$ CREATENODE$(\mathcal{D}^{(0)}, \mathcal{C}^{(0)}, \phi^{(0)}, \mathcal{S}^{(0)}, \mathcal{I}^{(0)})$
7: TREE $\leftarrow \{v_{\text{root}}\}$
8: $v_{\text{current}} \leftarrow v_{\text{root}}$
9: $\pi \leftarrow$ REVERSETOPOLOGICALORDER$(G)$         ▷ Fixed processing order
10: **// Main search loop**
11: **while** budget not exhausted **do**
12:   $e \leftarrow$ SELECTENTITY$(v_{\text{current}}, \pi)$          ▷ Algorithm 2
13:   **if** $e = $ NULL **then**
14:    **break**              ▷ All entities processed
15:   **// Inner level: parallel beam search over $W$ error batches**
16:   $\{\mathbf{err}_1, \ldots, \mathbf{err}_W\} \leftarrow$ SAMPLEERRORBATCHES$(e, v_{\text{current}}, W)$
17:   $\{v'_1, \ldots, v'_W\} \leftarrow$ EXPANDPARALLEL$(v_{\text{current}}, e, \{\mathbf{err}_1, \ldots, \mathbf{err}_W\})$    ▷ Alg. 3
18:   TREE $\leftarrow$ TREE $\cup \{v'_1, \ldots, v'_W\}$
19:   **// Select first worthy candidate**
20:   $v^* \leftarrow$ NULL
21:   **for** $v' \in \{v'_1, \ldots, v'_W\}$ **do**
22:    **if** ISWORTHY$(v', v_{\text{current}}, e)$ **then**
23:     $v^* \leftarrow v'$
24:     **break**
25:   **// Commit or mark intractable**
26:   **if** $v^* \neq$ NULL **then**
27:    $\mathcal{S}_{v^*} \leftarrow \mathcal{S}_{v_{\text{current}}} \cup \{e\}$
28:    $v_{\text{current}} \leftarrow v^*$
29:   **else**
30:    $\mathcal{I}_{v_{\text{current}}} \leftarrow \mathcal{I}_{v_{\text{current}}} \cup \{e\}$          ▷ Mark intractable
31: **// Return best solution found**
32: **return** $\mathcal{D}_{v_{\text{current}}}$

---

---

**Algorithm 2** SELECTENTITY: Reverse Topological Selection with Error Priority

---

**Require:** Node $v$, topological order $\pi$
**Ensure:** Selected entity $e$ or NULL
1: $\mathcal{U} \leftarrow \mathcal{E} \setminus (\mathcal{S}_v \cup \mathcal{I}_v)$          $\triangleright$ Unsolved entities
2: $\mathcal{C} \leftarrow \emptyset$          $\triangleright$ Eligible candidates
3: **// Collect entities whose dependencies are resolved**
4: **for** $e$ in $\pi$ **do**          $\triangleright$ Iterate in reverse topological order
5:     **if** $e \in \mathcal{U}$ **and** CALLEES$(e) \subseteq \mathcal{S}_v \cup \mathcal{I}_v$ **then**
6:         $\mathcal{C} \leftarrow \mathcal{C} \cup \{e\}$
7: **if** $\mathcal{C} = \emptyset$ **then**
8:     **return** NULL
9: **// Prioritize by error count (fewer errors first)**
10: **return** $\arg\min_{e \in \mathcal{C}} |\mathcal{F}_e^{(v)}|$          $\triangleright$ $\mathcal{F}_e^{(v)}$: failing tests for $e$ at node $v$

---

---

**Algorithm 3** EXPANDPARALLEL: Parallel Node Expansion

---

**Require:** Parent node $v$, target entity $e$, error batches $\{\mathbf{err}_1, \ldots, \mathbf{err}_W\}$
**Ensure:** Child nodes $\{v'_1, \ldots, v'_W\}$
1: **for** $k = 1$ to $W$ **in parallel do**
2:     **// Error-driven refinement (two-stage)**
3:     $f_k \leftarrow$ DIAGNOSE$(d_e^{(v)}, \mathbf{err}_k)$          $\triangleright$ Stage 1: Diagnosis
4:     $d_e^{(k)} \leftarrow$ PRESCRIBE$(d_e^{(v)}, f_k)$          $\triangleright$ Stage 2: Prescription
5:     $\mathcal{D}^{(k)} \leftarrow \mathcal{D}^{(v)}[e \mapsto d_e^{(k)}]$          $\triangleright$ Update documentation
6:     **// Regenerate and evaluate**
7:     $\mathcal{C}^{(k)} \leftarrow$ GENERATE$(\mathcal{D}^{(k)})$
8:     $\phi^{(k)} \leftarrow$ EVALUATE$(\mathcal{C}^{(k)}, \mathcal{T})$
9:     **// Create child node**
10:     $v'_k \leftarrow$ CREATENODE$(\mathcal{D}^{(k)}, \mathcal{C}^{(k)}, \phi^{(k)}, \mathcal{S}_v, \mathcal{I}_v)$
11:     SETPARENT$(v'_k, v)$
12: **return** $\{v'_1, \ldots, v'_W\}$

---

## C. Human Evaluation Rubric

Annotators rated each documentation on a 1–5 Likert scale across five dimensions, from the perspective of a human reader trying to understand the entity's purpose and behavior. Annotators had access to both the documentation under review and the reference source code; the source code served as ground truth for factual checks (e.g., Correctness, Completeness), while the rating reflected the documentation's quality as a reading aid. The rubric below was provided to all annotators prior to rating, together with two calibration examples (one high-quality, one low-quality), and was the only source of guidance on dimension semantics. Annotators were instructed to (i) rate dimensions independently to mitigate halo effects, and (ii) avoid central tendency by using the full 1–5 range when warranted.

**Completeness** — coverage of the entity's documented surface.

- Diagnostic checklist: (a) purpose / one-line summary; (b) each parameter's role and accepted values; (c) return value and its structure; (d) conditional behavior (branches, modes); (e) side effects, raised exceptions, or state mutation; (f) noteworthy edge cases (empty inputs, boundaries).

- **5**: Every checklist item that applies to the entity is addressed; the reader has no obvious unanswered "what does it do when..." question.

- **3**: Purpose and main parameters/return are covered, but two or more secondary items (e.g., edge cases, raised exceptions) are omitted.

- **1**: Multiple core items are missing; the reader cannot form a basic picture of inputs, outputs, or purpose.

**Correctness** — factual fidelity to the reference implementation.

- Diagnostic checklist: (a) types and shapes of inputs/outputs; (b) ordering, sorting, or sequencing claims; (c) conditional logic ("if X, then Y"); (d) handling of boundary values; (e) absence of fabricated parameters, fields, or behaviors not present in the code.

- **5**: All claims, including subtle ones (types, ordering, boundaries), are consistent with the reference implementation.

- **3**: Predominantly accurate, with one or two minor misstatements (e.g., a slightly wrong type, an under-stated boundary condition) that do not invert the overall meaning.

- **1**: Contains at least one major inaccuracy (e.g., wrong return semantics, fabricated parameter, inverted condition) that would mislead the reader about what the entity does.

**Clarity** — linguistic readability and unambiguity.

- Diagnostic checklist: (a) sentences parse on a single read; (b) no undefined jargon, project-internal terms, or unresolved acronyms; (c) pronouns and references have a single clear antecedent; (d) quantifiers and conditions are unambiguous (no "some", "usually", "if needed" without specification); (e) consistent terminology for the same concept.

- **5**: Reads fluently; every sentence admits a single reasonable interpretation.

- **3**: Mostly clear, but the reader pauses on one or two phrases that admit multiple readings or rely on undefined terms.

- **1**: Frequently ambiguous, awkwardly phrased, or self-contradictory; the reader cannot pin down a single intended meaning for core claims.

**Helpfulness** — utility as a reading aid for understanding the entity.

- Diagnostic checklist: (a) the documentation conveys what the entity does beyond restating its name or signature; (b) it surfaces non-obvious behavior that would otherwise require careful reading of the source; (c) it helps the reader anticipate output behavior for typical inputs; (d) it clarifies when this entity should be used vs. similarly-named alternatives.

- **5**: The documentation substantially accelerates understanding—a reader grasps the entity's behavior much faster than by reading code alone.

- **3**: The documentation provides useful orientation but the reader still relies heavily on the source for non-trivial details.

- **1**: The documentation adds little beyond what the signature already conveys; the reader gains almost no acceleration from reading it.

**Specificity** — level of operational detail ("how" vs "what").

- Diagnostic checklist: (a) explicit decision rules ("if X then Y, else Z") rather than vague verbs ("processes", "handles"); (b) concrete input–output relationships rather than abstract category labels; (c) named conditions, thresholds, or invariants where applicable; (d) ordering and iteration discipline made explicit when relevant.

- **5**: The documentation conveys the entity's logic at a mechanism level: rules, branches, and orderings are named explicitly.

- **3**: Mixes operational detail with high-level phrases; some behavior is specified concretely, other behavior is only labeled ("handles edge cases", "returns the result").

- **1**: High-level task labeling only (e.g., "processes the input and returns a result"); the reader is told *what* the entity is for but not *how* it produces its output.

## D. Documentation Showcase: Good vs. Bad

To illustrate the qualitative difference between partially and fully optimized documentation, we trace the refinement trajectory of a representative entity: `LineBreak.get_ohlc_data` from the `stocktrends` module. This entity generates a Line Break financial chart from price data—a task requiring precise specification of trend detection and reversal logic.

**D.1. Initial Documentation ($\phi = 0.25$)**

> **Initial Documentation**
>
> ```
> LineBreak.get_ohlc_data --- Generates a Line Break chart from input price data.
> Returns a DataFrame representing the chart.
> ```

The initial documentation states the purpose but provides no actionable detail. An LLM conditioned on this description has no way to infer the output columns, trend-tracking logic, or reversal handling.

**D.2. Refinement A: Partial Improvement ($\phi : 0.25 \rightarrow 0.50$)**

> **Refinement A (Partial)**
>
> ```
> LineBreak.get_ohlc_data --- Generates a Line Break chart as a DataFrame.  Starts with
> a number of initial lines equal to line_number and expands the chart based on price
> reversals and trend continuation.  Tracks uptrend or downtrend reversal conditions
> using the uptrend_reversal and downtrend_reversal methods.  Returns DataFrame with
> columns:  index, date, open, high, low, close, uptrend.
> ```

Refinement A adds output column names and references the helper methods, resolving half the failures. However, this is the failure mode of *partially-optimized* documentation: it specifies *what* the function returns but not *how* to produce it. The LLM can construct the DataFrame schema but cannot infer the actual OHLC value assignments.

**D.3. Refinement B: Fully Resolved ($\phi : 0.50 \rightarrow 1.00$)**

> **Refinement B (Fully Resolved)**
>
> ```
> LineBreak.get_ohlc_data --- Generates a Line Break chart by constructing a chart
> DataFrame from the input price data using trend continuation and reversal logic.
> Initializes the chart with the first line_number rows, setting all entries as
> uptrend.  Determines the current trend direction by comparing the last closing
> price to the first opening price.  In an uptrend, appends a new row when the current
> close exceeds the previous close, using the previous close as the open and low; in
> a downtrend, appends when the current close falls below the previous close, using
> the previous close as the open and high.  When a reversal is detected via the
> reversal-checking methods, flips the trend direction and assigns open/high/low/close
> values symmetrically.  Returns the constructed chart DataFrame.
> ```

Refinement B resolves all remaining failures by specifying (i) the exact trend detection rule (compare last close to first open), (ii) the precise OHLC assignment under each trend direction, and (iii) the reversal handling logic. The documentation now contains the full algorithmic specification required for an LLM to produce correct code.

**D.4. Key Observation**

This example illustrates the core principle that DOCSEARCH exploits: agent-oriented documentation quality is not a smooth function of detail level. Refinement A is substantially more detailed than the initial version yet only resolves half the test cases; the gap between Refinement A and Refinement B is precisely the inclusion of *operational* specifications—the exact computational rules—rather than *descriptive* ones. This is the qualitative pattern that human annotators rate as "Specificity" improvements in §4.5, and it is also the pattern that test-driven feedback most efficiently surfaces: failures on specific OHLC values directly reveal the missing assignment rules, while a human-readability metric would have no way to distinguish Refinement A from Refinement B.

# E. Prompt Templates

The error-driven refinement process consists of two stages: diagnosis and prescription.

## E.1. Stage 1: Diagnosis Prompt

---

Diagnosis Prompt Template

```
You are analyzing test failures to diagnose why an LLM failed to generate correct
code from a given documentation.
Entity Information:
- Name: {entity_name}
- Type: {entity_type} (function/class/method)
- Current Documentation:
{current_doc}
Generated Code (produced by the LLM from the current documentation; this is the
artifact under diagnosis):
{generated_code}
Test Failures:
The generated code failed the following tests:
{error_messages}
Task:
By comparing the generated code against the test failures and the current
documentation, diagnose:

 1. What went wrong? Identify the specific errors (type mismatches, incorrect
    outputs, missing logic, etc.)  visible in the generated code.

 2. Why did the LLM fail? What information is missing or ambiguous in the
    documentation that led to this implementation?

 3. What knowledge gap exists?  What does the documentation need to convey so that
    an LLM would not make this same mistake?

Output:
Provide a concise diagnosis in the following format:
- Error Type:  [brief description]
- Root Cause:  [what the documentation failed to convey]
- Missing Information:  [what needs to be clarified in the documentation]
```

---

## E.2. Stage 2: Prescription Prompt

---

Prescription Prompt Template

```
You are refining documentation for a code entity based on a diagnosis of test
failures.
Entity Information:
- Name:  {entity_name}
- Type:  {entity_type}
- Current Documentation:
{current_doc}
Diagnosis:
{diagnosis_output}
Task:
Refine the documentation to address the diagnosed issues.  The refined documentation
should:

 1. Directly address the identified knowledge gap.

 2. Clarify the ambiguous or missing information using general behavioral rules,
    invariants, or input--output relationships.

 3. Be precise enough for an LLM to generate correct code.

 4. Preserve all existing correct information.

Critical Constraint:
Do NOT hard-code specific test input values or expected output values from the
```

---

```
failures into the documentation.  The refined documentation must describe the
entity's general behavior, not memorize the particular test cases used during the
search.
Output:
Provide only the refined documentation, without any explanation.
```

### E.3. Code Generation Prompt

Code Generation Prompt Template

```
Generate Python code for all entities in this module based on their documentation.
Module:  {module_name}
Documentation:
{all_entity_documentation}
Requirements:

  • Implement each entity exactly according to its documentation

  • Ensure all cross-entity calls are consistent

  • Do not add functionality not specified in the documentation

Output:
Provide the complete Python implementation for all entities, without any explanation
or markdown.
```

### E.4. Test Case Generation Prompt

This prompt elicits diverse test inputs from the LLM, which are then executed on the reference implementation to derive expected outputs (Appendix H). The prompt emphasizes coverage across normal, edge, and corner cases.

Test Case Generation Prompt Template

```
You are generating pytest-style test functions for a code entity.  Each test must be
a complete, self-contained function with concrete inputs and assertions.
Entity Information:
- Name:  {entity_name}
- Type:  {entity_type} (function/class/method)
- Signature:  {signature}
- Source Code:
{source_code}
Dependencies:
The following entities are called by this entity:
{dependency_signatures}
Coverage Feedback (uncovered lines):
{uncovered_lines_with_source_snippets}
External Dependencies Detected:
{detected_io_network_time_calls}
Task:
Generate diverse pytest test functions that cover:

 1. Normal cases:  typical usage scenarios

 2. Edge cases:  boundary conditions, empty inputs, single elements

 3. Corner cases:  special values, type boundaries

 4. Targeted cases:  inputs that exercise the listed uncovered lines

Requirements:

  • Each test must be a complete def test_*(): function with concrete inputs and at
    least one assert statement
```

- Include informative assertion messages where possible (e.g., assert x == y,
  f"Expected {y} got {x}")

- For any detected external dependency, isolate it with a pytest fixture (mock
  filesystem/network, freeze time, seed randomness)

- Each test must directly exercise the target entity, not reach it only through
  unrelated callers

**Output:**
Provide a Python module containing only the test functions and any required fixtures,
without any explanation or markdown.

## E.5. Iterative Refinement Prompt (Baseline)

Iterative Refinement Prompt Template

```
You are refining documentation for a code entity based on test failures.
Entity Information:
- Name: {entity_name}
- Type: {entity_type}
- Current Documentation:
{current_doc}
Generated Code (produced by the LLM from the current documentation):
{generated_code}
Test Failures:
The generated code failed the following tests:
{all_error_messages}
Task:
Refine the documentation to fix the issues revealed by the test failures.  The
refined documentation should enable an LLM to generate correct code.  Do not
hard-code specific test input or output values into the documentation; describe
behavior in general terms.
Output:
Provide only the refined documentation, without any explanation.
```

**Note:** Unlike DOCSEARCH's two-stage diagnosis-prescription process, the iterative baseline conditions on *all* error messages simultaneously in a single refinement step, which can lead to the signal corruption phenomenon described in §3.3.

## E.6. Commercial Agent Prompts

We evaluate commercial coding agents (Cursor and Claude Code) on the task of generating documentation for code entities. Below are the prompt templates for each configuration.

### E.6.1. BASE CONFIGURATION PROMPT

Commercial Agent Base Prompt

```
Generate documentation for all entities in this module, refining it through
self-critique.  Your final output (documentation only) will be evaluated by an
external pipeline that generates code from your documentation and runs it against
held-out test cases.
Module: {module_name}
Entities (signatures and reference code):
{all_entity_signatures_and_code}
Internal Workflow (you may run up to B refinement iterations within your own
session):

 1. Draft documentation for each entity describing its purpose, parameters, return
    values, and behavior.
```

2. Critically review your own documentation:  identify any ambiguity, missing edge
   cases, or under-specified behavior that could lead another LLM (which does not
   see the source code) to generate an incorrect implementation.

3. Refine the documentation to address the issues identified in step 2.

4. Repeat steps 2--3 until you judge the documentation to be unambiguous and
   complete, or until the iteration budget $B$ is exhausted.  Each pass through steps
   2--3 counts as one iteration.

**Output:**
Provide only the final documentation for each entity, without source code or
explanation.

### E.6.2. +TOPO CONFIGURATION PROMPT

**Commercial Agent +Topo Prompt**

Generate documentation for all entities in this module, using the call-dependency
structure and refining through self-critique.  Your final output (documentation
only) will be evaluated by an external pipeline that generates code from your
documentation and runs it against held-out test cases.
**Module:**  {module_name}
**Dependency Structure (callee ← caller):**
{dependency_graph}
**Entities (signatures and reference code):**
{all_entity_signatures_and_code}
**Internal Workflow (you may run up to $B$ refinement iterations across all entities
within your own session):**

1. Process entities in reverse topological order (callees before callers), drafting
   documentation for each.

2. For the current entity, critically review your own documentation:  identify any
   ambiguity, missing edge cases, or under-specified behavior; also check that the
   documentation is consistent with the (already-stabilized) documentation of its
   callees.

3. Refine the documentation to address the issues identified in step 2.  Each
   refinement counts as one iteration against the budget $B$.

4. Move to the next entity in reverse topological order, and repeat steps 2--3.

5. Continue until all entities have been processed and you judge the documentation
   complete, or until the global budget $B$ is exhausted.

**Output:**
Provide only the final documentation for each entity, without source code or
explanation.

### E.6.3. +TEST CONFIGURATION PROMPT

**Commercial Agent +Test Prompt**

Generate documentation for all entities in this module, using a self-verification
loop.  Your final output (documentation only) will be evaluated by an external
pipeline that generates code from your documentation and runs it against held-out
test cases.
**Module:**  {module_name}
**Entities (signatures and reference code):**
{all_entity_signatures_and_code}
**Internal Workflow (you may iterate within your own session):**

```
 1. Draft documentation for each entity.

 2. Generate Python test cases that would distinguish a correct implementation
    of your documentation from common mis-implementations.  Validate these tests
    against the provided reference source code:  if a test fails on the reference,
    the test itself is incorrect and must be revised.

 3. Generate a candidate implementation from your documentation alone (i.e., without
    looking at the reference).

 4. Run the validated tests against this candidate implementation.  If any tests
    fail, the failure indicates that your documentation is ambiguous or incomplete
    --- refine the documentation accordingly.

 5. Repeat steps 3--4 until the candidate implementation passes all validated tests,
    or until your iteration budget is exhausted.
```

**Output:**
Provide only the final documentation for each entity, without source code, tests, or
explanation.

### E.6.4. +TOPO+TEST CONFIGURATION PROMPT

Commercial Agent +Topo+Test Prompt

Generate documentation for all entities in this module, using the call-dependency
structure and a self-verification loop.  Your final output (documentation only) will
be evaluated by an external pipeline that generates code from your documentation and
runs it against held-out test cases.
**Module:**  {module_name}
**Dependency Structure (callee ← caller):**
{dependency_graph}
**Entities (signatures and reference code):**
{all_entity_signatures_and_code}
**Internal Workflow (you may run up to $B$ refinement iterations across all entities
within your own session):**

```
 1. Process entities in reverse topological order (callees before callers), drafting
    documentation for each.

 2. For the current entity, generate Python test cases that would distinguish
    correct implementations from common mis-implementations; validate them against
    the reference source code and discard any test that the reference fails.

 3. Generate a candidate implementation of the current entity from its documentation
    alone (no reference).  Run the validated tests; if any fail, refine the
    documentation.  Each refinement of an entity's documentation counts as one
    iteration against the budget $B$.

 4. When moving to a caller, ensure its documentation is consistent with the
    (already-stabilized) documentation of its callees.

 5. Repeat until all entities pass their self-verification, or the global iteration
    budget $B$ is exhausted.
```

**Output:**
Provide only the final documentation for each entity, without source code, tests, or
explanation.

## F. Baseline Implementation Details

**Static Methods.  RepoAgent** (Luo et al., 2024): We use the official implementation with default hyperparameters.
Documentation is generated in a single pass without test feedback. **DocAgent** (Yang et al., 2025): We use the official

implementation. Multi-agent collaboration generates documentation iteratively until convergence.

**Commercial Agents.**  We evaluate **Cursor** and **Claude Code** under four configurations: (1) *Base*: generate documentation for all entities in a single session; (2) *+Topo*: provide the dependency graph as context for one-shot documentation generation; (3) *+Test*: prompt the agent to generate test cases from documentation, then iteratively refine code until tests pass, with a budget of 50 manual action approvals (agents typically terminate before exhausting the budget); (4) *+Topo+Test*: combine both strategies. Cursor supports Gemini-2.5-Flash and Claude-4.5-Sonnet but not GPT-4o; Claude Code only supports Claude models. Prompts are provided in Appendix E.6.

**Iterative Refinement.**  Algorithm 4 presents the iterative baseline. The **Iterative** variant selects entities randomly, while **Iterative+Topo** follows reverse topological order. Neither variant employs the worthy condition, allowing regressions.

---

**Algorithm 4** Iterative Refinement Baseline

---

**Require:** Documentation $\mathcal{D}$, max iterations $T$
1: **for** $t = 1$ to $T$ **do**
2:     $\mathcal{F} \leftarrow$ entities with failing tests
3:     $e \leftarrow$ SELECTENTITY($\mathcal{F}$)                                    ▷ Random or Topo order
4:     $\mathbf{e} \leftarrow$ error messages for $e$
5:     $d'_e \leftarrow$ REFINE($d_e, \mathbf{e}$)
6:     $\mathcal{D} \leftarrow \mathcal{D}[e \mapsto d'_e]$

---

# G. DevEval+ Benchmark

This appendix provides details about the evaluation suite used in our experiments, intended to enable reproducibility rather than to introduce a new benchmark.

## G.1. Composition

The suite comprises **20 Python modules** with **227 entities** (functions, classes, and methods) validated by **631 test cases**. Each module contains 11.4 entities on average, with 2.8 tests per entity.

*Table 5.* Modules used in our evaluation suite. **Source**: D = DevEval, G = GitHub.

| Module | Source | Domain |
|---|---|---|
| arXiv_digest | D | Academic/NLP |
| hybrid_images | D | Image Processing |
| textcnn | D | Deep Learning/NLP |
| chakin | D | NLP/Embeddings |
| geotext | D | Geography/NLP |
| hone | D | Data Processing |
| lice | D | Utilities |
| particle-swarm-optimization | D | Optimization |
| readtime | D | Text Processing |
| stocktrends | D | Finance |
| taskflow | G | Workflow/Systems |
| quantfin | G | Quantitative Finance |
| toon | G | Image Processing |
| biomarker-analysis | G | Bioinformatics |
| streamprocess | G | Data Streaming |
| cronparser | G | Scheduling/Cron |
| graphkit | G | Graph Algorithms |
| signalflow | G | Signal Processing |
| policyengine | G | Rule Engine |
| datavalidator | G | Data Validation |

**Selection Criteria.** Modules were selected to satisfy three criteria relevant to studying output coupling: (i) call-graph depth of at least three, so that coupling effects manifest non-trivially; (ii) every entity admits well-defined input–output behavior amenable to automated testing; and (iii) minimal external dependencies beyond the Python standard library, to keep evaluation reproducible.

**Sources.** The suite combines 10 modules from DevEval (Li et al., 2025)—whose original human-verified test cases we retained without modification—and 10 additional modules from open-source Python repositories under permissive licenses (MIT, Apache 2.0, BSD), each with an existing test suite. For the GitHub modules, we executed the original test suite against the reference source code and retained only the tests that passed, ensuring every test serves as a reliable oracle of intended behavior. The combined suite spans NLP, image processing, finance, optimization, bioinformatics, and signal processing.

### G.2. Dependency Structure

The suite exhibits the structural complexity needed to induce meaningful output coupling effects: across all 20 modules, the average call-graph depth is 4.2 (maximum 9). This depth ensures that refinements to callee documentation can propagate through multiple levels of callers, creating the conditions necessary to observe and address the whack-a-mole phenomenon.

### G.3. Test Suite

All tests are sourced from existing ground-truth suites (DevEval's human-verified tests and the original repositories' test suites). Each retained test was verified to (i) pass on the reference implementation, and (ii) directly exercise its target entity rather than reach it only via transitive calls, so that failures can be localized to the entity under refinement.

### G.4. Cross-Language Subset

To assess cross-language transfer (§4.4), we translated a subset of the suite from Python to Java and Go. We selected modules whose semantics are amenable to cross-language transfer—i.e., those that do not rely on Python libraries lacking apple-to-apple counterparts in Java or Go. Translation preserved test inputs exactly, with Python data structures mapped to their target-language equivalents; expected outputs were adapted to the target type system while preserving semantic equivalence, and the surrounding harness code was rewritten in JUnit 5 for Java and the standard `testing` package for Go. This procedure preserves the input–output contract of original test, so passing a translated test is equivalent to passing its Python counterpart, allowing target-language code generation to be evaluated against the same behavioral specification as in the Python setting.

## H. Test Case Generation

This appendix describes the LLM-based test case generation process used to guide DOCSEARCH's search without access to ground-truth tests.

### H.1. Overview

DOCSEARCH operates as a self-improving framework where the search is guided by *LLM-generated* test cases rather than ground-truth tests. This design choice reflects realistic deployment scenarios where comprehensive test suites may not exist. The key insight is that even imperfect tests provide valuable diagnostic signals—test failures reveal knowledge gaps in the documentation, enabling targeted refinement.

The test generation process follows a generate-then-validate approach:

1. **Test generation**: The LLM generates complete test functions—including inputs, assertions, and error messages—based on entity signatures and reference source code.

2. **Validation**: Each generated test is executed on the reference implementation. Only tests that pass are retained; those that fail (due to incorrect assertions or invalid inputs) are discarded.

Two additional filters are applied post-validation: (i) tests that reach the target entity only via transitive calls to other functions are discarded, ensuring each retained test directly exercises its target; and (ii) tests with identical normalized inputs

and assertions are deduplicated. This ensures test correctness by construction: every retained test is consistent with the reference implementation and contributes a distinct diagnostic signal.

## H.2. Coverage-Driven Test Generation

For each entity $e_i$, we employ a coverage-driven generation strategy in which the LLM generates test functions iteratively until branch coverage exceeds a target threshold $\tau_{\text{cov}}$. Each iteration asks the LLM to cover a mix of normal usage scenarios, edge cases such as empty inputs or single-element collections, corner cases involving special values that exercise specific code paths, and—once coverage feedback is available—targeted inputs aimed at the branches still left uncovered.

The generation loop terminates when (1) branch coverage reaches $\tau_{\text{cov}}$ (primary criterion), or (2) the per-iteration branch coverage gain stays below a positive threshold $\varepsilon$ for $P$ consecutive iterations (no-progress early-stop). The loop is bounded because (i) progress iterations (gain $\geq \varepsilon$) number at most $\lfloor 1/\varepsilon \rfloor$, since branch coverage is bounded above by 1.0 and each contributes at least $\varepsilon$; and (ii) no-progress iterations (gain $< \varepsilon$) cannot accumulate indefinitely because the $P$-th consecutive one triggers the early-stop.

The generation prompt (provided in Appendix E) includes the entity's signature and type information, the reference source code so that the LLM can infer the intended behavior, the signatures of its callees to keep generated tests consistent with downstream dependencies, and—when available—coverage feedback in the form of (line number, source snippet) pairs extracted from `coverage.py`'s branch-coverage output, indicating which lines remain uncovered.

## H.3. Quality Considerations

While the validation step ensures test correctness, LLM-generated tests may have quality limitations that motivate our diversified error batch sampling (§3.3).

**Error Message Quality Variation.** Since the LLM writes complete test functions, the diagnostic value of error messages depends entirely on how each assertion is written. Some tests include informative messages (e.g., `assert result == [1,2,3], f"Expected [1,2,3] got {result}"`) that directly reveal the failure mode, while others use bare assertions (e.g., `assert result == expected`) that produce only `AssertionError` with no actionable signal. This heterogeneity is inherent to LLM-generated tests and cannot be controlled a priori.

**Input Quality Variation.** Not all generated test inputs are equally informative. Some inputs may exercise redundant code paths, miss critical edge cases, or cluster around similar patterns. By conditioning refinements on different test subsets, diversified error batch sampling increases the probability of encountering informative failure signals.

**Cascading Failures.** When entity $e_i$ calls entity $e_j$, an incorrect implementation of $e_j$ can cause $e_i$'s tests to fail even if $e_i$'s documentation is correct. These cascading failures are *noisy* in the sense of Definition 4—they do not reveal $e_i$'s knowledge gap. Our reverse topological ordering addresses this by ensuring callees are finalized before callers.

**External Dependencies.** External dependencies (filesystem, network, time, randomness) are detected via static analysis of import statements and AST patterns; the LLM is then prompted to isolate them through `pytest` fixtures (mocks for filesystem and network calls, frozen-time wrappers for time-dependent code) so that test outcomes remain deterministic and reproducible across runs.

## H.4. Hyperparameters

We use the following hyperparameters for test generation:

- **LLM**: GPT-4o (fixed across all experiments for consistency)

- **Temperature**: $\tau = 0.7$ for input diversity

- **Coverage threshold**: $\tau_{\text{cov}} = 90\%$ branch coverage

- **No-progress patience**: $P = 2$ consecutive iterations within $\varepsilon = 0.01$ coverage gain

# I. Additional Related Work: Dependency-Guided LLM Generation

A growing line of work uses structural dependencies—ASTs, call graphs, repository dependency graphs, or dataflow—to guide LLM-based generation. For repository-level code completion, graph- and structure-based retrieval methods scope the LLM's context (Shrivastava et al., 2023; Zhang et al., 2023a; Wu et al., 2024; Liu et al., 2025a). At the edit and agent level, dependency structure drives plan propagation, context plug-ins, AST-aware search APIs, and decoder-level constraints from static analysis (Bairi et al., 2024; Ouyang et al., 2025; Zhang et al., 2024; Agrawal et al., 2023), exemplified by CodePlan (Bairi et al., 2024), which maintains a repository-wide dependency graph and propagates edits along a plan graph derived from incremental dependency analysis. For repository-level documentation, RepoAgent (Luo et al., 2024) and DocAgent (Yang et al., 2025) use global structure analysis and topological traversal, with DocAgent in particular processing entities in topological order so each is documented after its dependencies. Beyond code, NL2LOGIC (Putra et al., 2026) uses an AST as an intermediate representation for translating natural language into first-order logic. DOCSEARCH walks along this path and uses the call-dependency DAG to govern a *search* over the documentation space, where test-time feedback drives iterative refinements and a worthy condition prevents the regressions induced by output coupling.

