# OpenReview forum: "Escaping Whack-a-Mole: Optimizing Documentation as Repo-Specific Playbooks for Coding Agents"
_ICML.cc/2026/Conference — ICML 2026 regular_

### Official Review · Reviewer_YWRP · 2026-02-23

**Soundness:** 2
**Presentation:** 4
**Significance:** 2
**Originality:** 2
**Overall Recommendation:** 4
**Confidence:** 4

**Summary:**

This paper optimizes code documentation so an LLM can implement dependency-coupled modules using only the docs and pass tests. It identifies “whack-a-mole” failures from output coupling, and proposes DOCSEARCH, a dependency-guided bi-level search that refines docs in reverse topological order with test-driven feedback and a non-regression (“worthy”) acceptance rule. On DevEval+, DOCSEARCH outperforms static docs and iterative self-refinement across multiple LLMs and also improves cross-language generation.

**Compliance With Llm Reviewing Policy:**

Affirmed.

**Final Justification:**

I find the paper technically sound, clearly presented, and meaningful in its problem formulation, while the rebuttal substantially addressed my main concerns and improved my assessment. I am therefore updating my recommendation to weak accept, though I think this remains a borderline case and I would not have strong objections if it is ultimately not accepted.

**Key Questions For Authors:**

1. How stable are the LLM-generated tests and DOCSEARCH outcomes across different prompts/temperatures/seeds (please report mean±std over multiple runs)?
2. What are the main failure modes of DOCSEARCH versus the baselines (e.g., test-generation errors, search-path drift, dependency regressions), and how frequently does each occur?
3. How do you ensure generated tests use correct APIs and valid inputs/types at module/repo scale, and what is the observed test-generation failure rate?
4. What is the cost or DOCSEARCH and each baseline, and how does cost scale with dependency-graph size/depth?

**Limitations:**

yes

**Strengths And Weaknesses:**

**Strengths**:
1. The paper crisply formulates agent-oriented documentation optimization and pinpoints output coupling along dependency edges as the root cause of “whack-a-mole” oscillations in naive iterative refinement.
2. DOCSEARCH is well-motivated and structured—dependency-guided (reverse-topological) outer scheduling plus a bi-level search with a non-regression (“worthy”) acceptance rule that promotes monotonic progress.
3. On DevEval+, DOCSEARCH shows consistent, sizable gains over static documentation and iterative self-refinement across multiple LLM backbones, with an additional encouraging cross-language transfer signal.

**Weakness**:
1. DOCSEARCH is guided by LLM-generated tests, but the test generator is itself conditioned on the current documentation—exactly what the algorithm keeps modifying—creating a potentially circular and unstable feedback loop. Early imprecise docs can produce weak or biased tests that steer edits toward “fitting” those tests rather than the true specification. Moreover, repo-/module-level test generation requires not only correct assertions but also correct API usage and valid inputs/types; the paper does not clearly explain how this is ensured or how often test generation fails, raising the risk of negative optimization or reward hacking (passing flawed generated tests while hurting ground-truth performance).
2. The pipeline compounds multiple LLM stochastic components (test generation, code generation, doc-edit proposal/search), making outcomes likely sensitive. It also lacks a failure-mode breakdown quantifying how much improvement is attributable to the method relative to baselines.
3. DevEval+ remains a module-level benchmark. It is unclear whether the claimed “escaping whack-a-mole” behavior holds for larger, messier repositories with multi-package boundaries, configuration-driven behavior, dynamic dispatch/reflection, hidden side effects, or cross-language/cross-process interfaces. The current evaluation does not provide sufficient evidence that the proposed dependency-guided scheduling and non-regression criteria will scale or remain effective in these more realistic settings.
4. DOCSEARCH is inherently a bi-level search procedure with repeated doc proposals, code generations, and test executions, so practical viability depends critically on budget. The paper does not provide a thorough cost comparison to baselines. It is also unclear how cost grows with dependency-graph size/depth, which limits the ability to assess whether the approach is a practical optimization strategy or an expensive offline procedure.

---

> ### Author Rebuttal · Authors · 2026-03-31
>
> We thank the reviewer for the thorough and detailed review.
>
> ## Q1: Stability and Improvement Attribution
> **Stability.** DocSearch is robust to stochastic variation: across 5 random seeds on 6 modules, φ̄ = 93.4 ± 0.4% (vs. Iterative 57.2 ± 2.8%, Iter+Topo 61.0 ± 1.7%). Varying the documentation generation temperature across τ ∈ {0.3, 0.5, 0.7, 1.0} yields a narrow 2.1 pp range (φ̄: 91.2–93.3%). Full tables: [Appendix](https://anonymous.4open.science/r/docsearch_icml_rebuttal-2BF6/ywrp_stability.md).
>
> **Improvement attribution.** New ablation (GPT-4o, DevEval+) over the 36.7 pp gap from Iterative (56.6%): topo ordering +4.3 pp (→60.9%), worthy condition +7.1 pp (→68.0%), beam search +25.3 pp (→93.3%).
>
> ## Q2: Failure Mode Analysis
>
> **Observed failure modes (21/227 unsolved, 9.3%).** (1) *Specification error (7)*: correct documentation but LLM fails to follow instructions. (2) *Worthy deadlock (12)*: improving one entity regresses tightly coupled siblings; (3) *Unsearched (2)*: budget exhausted before the entity was reached.
>
> ## Q3: Test Generation Reliability and Circular Feedback
>
> Test generation is conditioned on the source code, not the documentation being optimized, so there is no circular dependency — tests remain fixed throughout the search. The LLM generates test inputs from reference code, ensuring correct API usage and valid input types; expected outputs are derived by executing these inputs on the reference implementation (§F.3, L1464–1480). Over 99% of generated inputs yield valid test oracles on DevEval+. However, some inputs may exercise redundant code paths or miss edge cases (§F.4, L1485–1494); this is addressed by our diversified error batch sampling: each beam branch conditions on a different test subset, reducing sensitivity to any individual weak test case (Theorem 3, §3.3).
>
> **New experiments on feedback signal (Table 1).** To directly address the concern about test quality, we conducted experiments varying the feedback signal. (1) *Signal source*: Replacing LLM-generated tests with ground-truth tests yields only 1.4 pp improvement, confirming that LLM-generated tests do not induce reward hacking. Replacing execution-based feedback with LLM-as-a-judge scoring drops φ̄ by 17.0 pp, showing that execution signals are essential and cannot be substituted by model judgment. (2) *Coverage reduction*: Reducing the coverage threshold τ\_cov from 0.9 to 0.3 degrades φ̄ by only 4.9 pp.
>
> **Table 1: Sensitivity to feedback signal (GPT-4o, DevEval+).**
> | Configuration | Feedback Signal | φ̄ (%) |
> |---|---|---|
> | DocSearch (default) | LLM tests | 93.3 |
> | Ground-truth tests | GT tests | 94.7 |
> | LLM-as-a-judge | Judge score | 76.3 |
> | τ\_cov = 0.7 | Reduced tests | 93.4 |
> | τ\_cov = 0.5 | Reduced tests | 92.8 |
> | τ\_cov = 0.3 | Reduced tests | 88.4 |
>
> ## Q4: Cost Analysis
>
> To enable fair comparison, we conducted new experiments varying DocSearch's beam budget B (Table 2). At B=20, DocSearch costs \\$25.3, comparable to Claude Code+Topo+Test (\\$24.9), yet achieves φ̄=81.0%, a +9.5 pp advantage. Scaling to B=50 further improves φ̄ to 93.3%. Token-budget analysis is shown in [Figure 1](https://anonymous.4open.science/r/docsearch_icml_rebuttal-2BF6/token_efficiency.png). Per-module cost is shaped by dependency depth, entity count, and per-entity difficulty; per-module breakdown in [Appendix](https://anonymous.4open.science/r/docsearch_icml_rebuttal-2BF6/cost_scaling.md).
>
> **Table 2: End-to-end cost (GPT-4o, DevEval+).**
>
> | Method | Cost (\\$) | φ̄ (%) |
> |---|---|---|
> | Iterative | 35.5 | 56.6 |
> | Iterative+Topo | 36.2 | 60.9 |
> | Cursor+Topo+Test | 13.8 | 59.4 |
> | Claude Code+Topo+Test | 24.9 | 71.5 |
> | DocSearch (B=20) | 25.3 | 81.0 |
> | DocSearch (B=30) | 32.9 | 87.0 |
> | DocSearch (B=50) | 48.2 | 93.3 |
>
> ## Weakness: Scalability Beyond Module-Level
>
> **Realistic complexity handling.** DocSearch's existing architecture only requires minor additions to address these cases. (1) *Cycles*: a preprocessing step contracts strongly connected components into meta-entities and refines them jointly, restoring a DAG for topological traversal. (2) *Imprecise call graphs* (dynamic dispatch, indirect calls, or missing edges): the worthy condition already detects regressions caused by missing edges. When refining entity B triggers regression on a previously solved entity A, a lightweight diagnostic step checks whether a dependency edge between them was missed by static analysis. If confirmed, the topological order is updated and the existing search proceeds as normal. We will add this discussion to the revised manuscript.
>
> We conducted new experiments on [SWE-Dev](https://arxiv.org/abs/2505.16975), a repo-level feature-implementation benchmark where agents build new functionality within existing codebases. We select 10 packages (67 instances, 35 unique source files). DocSearch+Source improves +14.8 pp pass and +11.7 pp solve over the PRD baseline, confirming that DocSearch remains effective on repo-level benchmark.

---

> > ### Author Rebuttal · Reviewer_YWRP · 2026-04-01
> >
> > Thank you for the detailed rebuttal and additional experiments. My main concerns have been sufficiently addressed, and I will raise my score from 3 to 4.

---

> > > ### Author Response · Authors · 2026-04-04
> > >
> > > Thank you for the detailed review and for raising your score. Your questions on stability, failure modes, and circular feedback led directly to some of our most informative analyses, including the multi-seed variance study, the failure mode taxonomy, and the ground-truth vs. LLM-test comparison confirming the absence of reward hacking. We are grateful that your thorough review gave us the opportunity to present a deeper and more rigorous characterization of DocSearch.

---

### Official Review · Reviewer_qPat · 2026-03-12

**Soundness:** 2
**Presentation:** 3
**Significance:** 2
**Originality:** 3
**Overall Recommendation:** 4
**Confidence:** 3

**Summary:**

This paper studies agent-oriented documentation generation, where documentation is optimized not for human readability but for the correctness of code generated by an LLM conditioned on that documentation. The core claim is that standard iterative refinement fails because of output coupling across program entities, creating a whack-a-mole effect where fixing one entity can break dependent ones. To address this, the authors propose DOCSEARCH, a bi-level framework with (i) reverse-topological, dependency-guided entity selection, (ii) diversity-augmented beam search over documentation refinements using sampled subsets of failing tests, and (iii) a worthy condition that commits only non-regressive refinements. Experiments on DEVEVAL+ show large gains over static documentation methods, commercial coding agents, and iterative refinement baselines, plus promising cross-language transfer from Python to Java.

**Compliance With Llm Reviewing Policy:**

Affirmed.

**Final Justification:**

The rebuttal addressed my main concerns.

**Key Questions For Authors:**

1. How sensitive is DOCSEARCH to the quality of the LLM-generated tests used during search? Can you report controlled analyses where pseudo-test quality or coverage is varied?
2. What is the full inference-time cost of DOCSEARCH relative to Iterative+Topo and commercial-agent baselines?
3. How exactly is the worthy condition implemented efficiently in practice?
4. The method relies on a dependency DAG and reverse topological traversal. How are cycles, dynamic dispatch, indirect calls, or incomplete call graphs handled in the benchmark and in the implementation?
5. Can cross-language documentation transfer generalize to more programming languages?
6. Is there a typo in the indicator function in Eq. (1)?

**Limitations:**

yes

**Strengths And Weaknesses:**

### Strengths
1. The paper articulates a compelling shift from human-oriented to agent-oriented documentation and formalizes the task as black-box optimization over documentation quality measured by downstream code correctness.
2. The decomposition into reverse-topological outer search, diversified inner beam search, and the worthy condition is conceptually coherent and directly aligned with the identified output-coupling problem.
3. On DEVEVAL+, DOCSEARCH substantially outperforms multiple baseline classes across Gemini-2.5-Flash, Claude-4.5-Sonnet, and GPT-4o, and the ablations and trajectory visualizations help support the claimed mechanism behind the gains.

### Weaknesses
1. Although final evaluation uses ground-truth tests, the optimization process relies on self-generated tests, so the method’s robustness to low-quality or systematically biased pseudo tests remains insufficiently characterized.
2. The method performs repeated code generation, testing, beam search, and candidate verification; however, the paper does not clearly quantify token/API cost, wall-clock runtime, or how performance scales with repository size, graph depth, and beam budget.
3. The worthy condition may be expensive in practice. Since a refinement is committed only if it causes no regression, the practical cost of checking worthiness across entities/tests could be substantial, but this trade-off is not fully analyzed.
4. The Python $\rightarrow$ Java transfer experiment is performed on only 10 modules due to manual test translation, which makes the broader claim of language-agnostic intermediate representation suggestive rather than fully established.

---

> ### Author Rebuttal · Authors · 2026-03-31
>
> We thank the reviewer for the thorough and rigorous review.
>
> ## Q1: Sensitivity to Pseudo-Test Quality
>
> We appreciate this meaningful question. Test oracles are derived by executing LLM-proposed inputs on the reference implementation to obtain expected outputs (§F.3, L1464–1480), so output correctness is guaranteed by construction. The remaining source of imperfection is input quality — some inputs may exercise redundant code paths or miss critical edge cases (§F.4, L1485–1494). However, this variation is precisely what our diversified error batch sampling is designed to handle: by conditioning each beam branch on a different test subset rather than all tests simultaneously, DocSearch reduces sensitivity to any individual weak or redundant test case (Theorem 3, §3.3).
>
> We conducted new controlled experiments varying the feedback signal along multiple dimensions (Table 1). (1) *Signal source*: Replacing LLM-generated tests with ground-truth tests yields only 1.4 pp improvement, confirming that LLM-generated tests do not induce reward hacking. Replacing execution-based feedback with LLM-as-a-judge scoring drops φ̄ by 17.0 pp, showing that execution signals are essential and cannot be substituted by model judgment. (2) *Coverage reduction*: Reducing the coverage threshold τ_cov from 0.9 to 0.3 degrades φ̄ by only 4.9 pp, demonstrating graceful degradation.
>
> **Table 1: Sensitivity to feedback signal (GPT-4o, DevEval+).**
> | Configuration | Feedback Signal | φ̄ (%) |
> |---|---|---|
> | DocSearch (default) | LLM tests | 93.3 |
> | Ground-truth tests | GT tests | 94.7 |
> | LLM-as-a-judge | Judge score | 76.3 |
> | τ_cov = 0.7 | Reduced tests | 93.4 |
> | τ_cov = 0.5 | Reduced tests | 92.8 |
> | τ_cov = 0.3 | Reduced tests | 88.4 |
>
> ## Q2&Q3: Inference-Time Cost and Worthy Condition Implementation
>
> Thank you for raising this important concern. Test generation is a one-time offline step; runtime cost comprises code generation → test execution → diagnosis → prescription. Code generation dominates runtime (67% of tokens); diagnosis 19%; prescription 14%. The worthy condition is programmatic (no LLM): after regenerating code for the refined entity, we re-run tests and compare pass rates against the previous state. This check is negligible relative to code generation.
>
> **Cost comparison (Table 2).** To enable fair comparison, we conducted new experiments varying DocSearch's beam budget B. At B=20, DocSearch costs \\$25.3, comparable to Claude Code+Topo+Test (\\$24.9), yet achieves φ̄=81.0%, a +9.5 pp advantage. Scaling to B=50 further improves φ̄ to 93.3%. Token-budget analysis further corroborates this: at just 20k output tokens, DocSearch (φ̄=68.3%) already surpasses Iterative's final performance (56.6% at 200k), a 10× efficiency gap ([Figure 1](https://anonymous.4open.science/r/docsearch_icml_rebuttal-2BF6/token_efficiency.png)). DocSearch completes in 499 minutes for all 20 modules, comparable to Iterative baselines (350–360 min).
>
> **Table 2: End-to-end cost (GPT-4o, DevEval+).**
>
> | Method | Cost (\\$) | φ̄ (%) |
> |---|---|---|
> | Iterative | 35.5 | 56.6 |
> | Iterative+Topo | 36.2 | 60.9 |
> | Cursor+Topo+Test | 13.8 | 59.4 |
> | Claude Code+Topo+Test | 24.9 | 71.5 |
> | DocSearch (B=20) | 25.3 | 81.0 |
> | DocSearch (B=30) | 32.9 | 87.0 |
> | DocSearch (B=50) | 48.2 | 93.3 |
>
> **Cost scaling.** Per-module cost is shaped collectively by depth, entity count, and per-entity difficulty. Full per-module breakdown: [Appendix](https://anonymous.4open.science/r/docsearch_icml_rebuttal-2BF6/cost_scaling.md).
>
> ## Q4: Cycles, Dynamic Dispatch, Incomplete Call Graphs
>
> We appreciate the opportunity to elaborate on this important aspect. DocSearch's existing architecture only requires minor additions to address these cases. (1) *Cycles*: a preprocessing step contracts strongly connected components into meta-entities and refines them jointly, restoring a DAG for topological traversal. (2) *Imprecise call graphs* (dynamic dispatch, indirect calls, or missing edges): the worthy condition already detects regressions caused by missing edges. When refining entity B triggers regression on a previously solved entity A, a lightweight diagnostic step checks whether a dependency edge between them was missed by static analysis. If confirmed, the topological order is updated and the existing search proceeds as normal.
>
> ## Q5: Cross-Language Generalization
> We expanded Python→Java from 10 to 16 modules (181 entities, 519 tests). Doc-only outperforms Source-only by 21.0 pp (φ̄: 71.4% vs. 50.4%); Doc+Source achieves 80.5%. We also conducted a new experiment on Python→Go (5 modules). Doc-only outperforms Source-only by 24.2 pp (φ̄: 68.9% vs. 44.7%); Doc+Source achieves 78.4%.
>
> ## Q6: Indicator Function in Eq. (1)
> Thank you for the careful reading — it is not a typo. The confusing rendering is due to a LaTeX issue (`\mathbb{1}` in amssymb produces a non-standard glyph). We will switch to `\mathds{1}` (dsfont) in the revision.

---

> > ### Author Rebuttal · Reviewer_qPat · 2026-04-03
> >
> > Thanks for the response and additional results. My concerns have been addressed, so I have increased my score from 3 to 4.

---

> > > ### Author Response · Authors · 2026-04-04
> > >
> > > Thank you for the rigorous review and for raising your score. Your insightful questions on pseudo-test sensitivity, cost structure, cycle handling, and cross-language generalization motivated us to conduct a series of controlled experiments that provide a richer and more thorough presentation of DocSearch's capabilities. We especially appreciate your attention to the dependency graph edge cases, which gave us the chance to articulate how DocSearch's existing architecture extends to handle these scenarios.

---

### Official Review · Reviewer_SR5r · 2026-03-13

**Soundness:** 4
**Presentation:** 4
**Significance:** 4
**Originality:** 4
**Overall Recommendation:** 6
**Confidence:** 4

**Summary:**

This paper proposes DOCSEARCH, a dependency-guided bi-level search framework that optimizes code documentation for LLM coding agents, treating documentation quality as a black-box objective measured by whether code generated from that documentation passes tests. The paper argues that existing documentation systems are mostly optimized for human readability or use simple iterative refinement, which breaks down for repository-level code because program entities are interdependent: improving one entity’s documentation can change its implementation and unintentionally break its callers, causing a “whack-a-mole” oscillation. To overcome this, the authors explicitly model dependency structure and only commit refinements that improve the current target without regressing already solved entities, so the search becomes monotonic rather than unstable. DOCSEARCH has two levels. At the outer level, it performs a priority search over the program’s dependency DAG in reverse topological order, so callees are stabilized before callers and downstream interference is reduced. At the inner level, for the selected entity, it runs a beam-style search over documentation refinements, where each branch conditions on a different sampled subset of failing tests so that the model sees more diverse diagnostic signals instead of overfitting to one noisy error bundle. Each refinement follows a diagnosis → prescription pipeline: first infer what knowledge gap in the documentation caused the bad code, then rewrite the documentation to fix that gap. A worthy condition connects both levels: a refinement is accepted only if it improves or preserves pass rates for all entities, which is what lets the method avoid the whack-a-mole failure mode and make monotonic progress. The evaluation is done on DEVEVAL+, a benchmark of 20 Python modules, 227 entities, and 631 test cases. DOCSEARCH outperformed existing works in terms of generating correct code from documentation that passed all the tests.

**Compliance With Llm Reviewing Policy:**

Affirmed.

**Final Justification:**

The rebuttal addressed all of my concerns.

**Key Questions For Authors:**

1. Can you provide showcases/ examples when the generated documentation is considered a good document and vice versa?
2. Can you do the analysis of how costly the search processes (outer level search, inner level search) perform?

**Limitations:**

Lack of evaluation of the aspect of documentation in terms of readability. I suggest that authors perform an evaluation of the generated documents by a human.

**Strengths And Weaknesses:**

Strength:
- The motivation of this work is clear and sound.
- Rigorous experiment on a well-known benchmark.
- Clear definition of important concepts like output coupling, which can be shown in Section 2.

Weakness:
- The experiment was measured by the correctness of the generated code only. A study that involved human for checking the quality of generated documents is needed.

---

> ### Author Rebuttal · Authors · 2026-03-31
>
> We thank the reviewer for the thoughtful review.
>
> ## Weakness: Human Evaluation of Documentation Quality
>
> We conducted a paired human evaluation comparing (A) initial documentation before test-driven refinement and (B) optimized documentation after all refinement cycles (GPT-4o, B=50). 30 entities from 6 modules where documentation changed. Rated by 5 CS-background annotators on five 1–5 Likert dimensions. We decompose the reviewer's suggested readability evaluation into five fine-grained dimensions of human-perceived documentation quality: Completeness, Correctness, Clarity, Helpfulness, and Specificity.
>
> **Table 1: Human evaluation results.** Mean ± std. α: Krippendorff's ordinal α.
>
> | Dimension | Initial | Optimized | α |
> |---|---|---|---|
> | Completeness | 3.10 ± 1.03 | 4.67 ± 0.54 | 0.94 |
> | Correctness | 3.58 ± 0.86 | 4.50 ± 0.85 | 0.78 |
> | Clarity | 2.88 ± 0.91 | 4.42 ± 0.67 | 0.92 |
> | Helpfulness | 2.40 ± 1.16 | 4.43 ± 0.80 | 0.71 |
> | Specificity | 2.17 ± 1.05 | 4.52 ± 0.85 | 0.75 |
>
> Optimized documentation improves across all five dimensions, with the largest gains in Specificity (+2.35) and Helpfulness (+2.03), followed by Completeness (+1.57) and Clarity (+1.54). Inter-annotator agreement is high (α ≥ 0.71) across all dimensions. We will include these results in the revised manuscript.
>
> ## Q1: Documentation Showcase
>
> We showcase good and bad documentation using `LineBreak.get_ohlc_data`.
>
> **Refinement A — partial improvement (φ: 0.25→0.50):**
> > LineBreak.get_ohlc_data — Generates a Line Break chart as a DataFrame. Starts with a number of initial lines equal to line_number and expands the chart based on price reversals and trend continuation. Tracks uptrend or downtrend reversal conditions using the uptrend_reversal and downtrend_reversal methods. Returns DataFrame with columns: index, date, open, high, low, close, uptrend.
>
> Refinement A added output column names and method references, resolving half the failures. However, the LLM still cannot infer *how* to compute each OHLC value — the documentation says *what* is returned but not *how* to produce it.
>
> **Refinement B — fully resolved (φ: 0.50→1.00):**
> > LineBreak.get_ohlc_data — Generates a Line Break chart by constructing a chart DataFrame from the input price data using trend continuation and reversal logic. Initializes the chart with the first line_number rows, setting all entries as uptrend. Determines the current trend direction by comparing the last closing price to the first opening price. In an uptrend, appends a new row when the current close exceeds the previous close, using the previous close as the open and low; in a downtrend, appends when the current close falls below the previous close, using the previous close as the open and high. When a reversal is detected via the reversal-checking methods, flips the trend direction and assigns open/high/low/close values symmetrically. Returns the constructed chart DataFrame.
>
> Refinement B resolved all remaining failures by specifying the exact trend condition and precise OHLC assignment rules for each case, enabling the LLM to directly translate them into code.
>
> More examples: https://anonymous.4open.science/r/docsearch_icml_rebuttal-2BF6/examples.md
>
> ## Q2: Cost of Search Processes
> **Outer-level search: negligible.** The priority search over the dependency DAG is entirely deterministic (no LLM): DAG construction via static AST (<1s), reverse topological ordering O(V+E), error-count priority selection. Total cost across DevEval+: <10 seconds.
>
> **Inner-level search: dominant cost.** The beam search accounts for all LLM cost. Each iteration comprises code generation → test execution → diagnosis → prescription → worthy condition check. As shown in Table 2, code generation accounts for 67% of total tokens (~204k/module), followed by diagnosis (19%, ~58k) and prescription (14%, ~43k).
>
> **Table 2: Token breakdown by component (GPT-4o, DevEval+).**
> | Component | Token % | Avg Tokens/Module |
> |---|---|---|
> | Code generation | 67% | ~204k |
> | Diagnosis | 19% | ~58k |
> | Doc refinement | 14% | ~43k |
>
> **Cost comparison (Table 3).** At matched budget ($25.3), DocSearch (B=20) achieves φ̄=81.0%, outperforming Claude Code+Topo+Test (71.5%) by +9.5 pp. Scaling to B=50 further improves φ̄ to 93.3%. Token-budget analysis further corroborates this: at just 20k output tokens, DocSearch (φ̄=68.3%) already surpasses Iterative's final performance (56.6% at 200k), a 10× efficiency gap ([Figure 1](https://anonymous.4open.science/r/docsearch_icml_rebuttal-2BF6/token_efficiency.png)). This is a one-time offline cost — optimized documentation is reused across all downstream tasks.
>
> **Table 3: End-to-end cost (GPT-4o, DevEval+).**
>
> | Method | Cost ($) | φ̄ (%) |
> |---|---|---|
> | Iterative | 35.5 | 56.6 |
> | Iterative+Topo | 36.2 | 60.9 |
> | Cursor+Topo+Test | 13.8 | 59.4 |
> | Claude Code+Topo+Test | 24.9 | 71.5 |
> | DocSearch (B=20) | 25.3 | 81.0 |
> | DocSearch (B=30) | 32.9 | 87.0 |
> | DocSearch (B=50) | 48.2 | 93.3 |

---

> > ### Author Rebuttal · Reviewer_SR5r · 2026-04-03
> >
> > The authors addressed all my concerns with rigorous analysis. I will raise my score for this paper.

---

> > > ### Author Response · Authors · 2026-04-04
> > >
> > > Thank you for the positive assessment and for raising your score. Your suggestion on human evaluation was particularly valuable. It motivated the five-dimension annotation study that reveals DocSearch's documentation improvements are not merely functional but also yield substantial gains in clarity, specificity, and helpfulness as perceived by human readers. We are grateful that your question gave us the opportunity to demonstrate this additional dimension of our work.

---

### Official Review · Reviewer_4z5r · 2026-03-17

**Soundness:** 3
**Presentation:** 3
**Significance:** 3
**Originality:** 3
**Overall Recommendation:** 4
**Confidence:** 3

**Summary:**

The paper addresses the "agent-oriented documentation generation" task, framing it as a black-box optimization problem. The authors identify "output coupling"—where refining one entity's documentation changes its implementation and breaks dependent callers—as the cause of a "whack-a-mole" oscillation in iterative refinement. They propose DOCSEARCH, a bi-level search framework that uses: (1) an outer-level priority search following reverse topological order to stabilize callees before callers; (2) an inner-level beam search with diversified error sampling to escape local optima ; and (3) a "worthy condition" to ensure monotonic progress and prevent regressions. On the DevEval+ benchmark, DOCSEARCH achieves a 90.7% solve rate, significantly outperforming commercial agents and iterative baselines.

**Compliance With Llm Reviewing Policy:**

Affirmed.

**Key Questions For Authors:**

1. Integration with Agent Skills: Have you explored integrating the DOCSEARCH methodology as a fixed "skill" or "tool" within a multi-agent framework (e.g., as a specialized documentation-optimization tool that an agent can call)? If so, how does it affect the agent's general reasoning efficiency?
2. Scalability to Repository-Level Benchmarks: DOCSEARCH is highly effective on DevEval+. Have you attempted to apply this framework to larger repository-level tasks like SWE-bench? Or do you think it is feasible for such tasks?

**Limitations:**

yes

**Strengths And Weaknesses:**

Strengths
1. Originality: The paper introduces a novel perspective by shifting documentation focus from human readability to "agent comprehension". The identification of "output coupling" as a fundamental barrier in LLM-based software engineering is insightful.
2. Significance: The discovery that optimized documentation acts as a language-agnostic intermediate representation for cross-language code generation (Python to Java) is highly significant for legacy modernization.
3. Presentation: The "whack-a-mole" analogy and the bi-level search visualization (Figure 2) effectively communicate complex dependency issues.

Weaknesses:
1. Computational Cost: The framework requires multiple code generation and execution cycles (beam search width $W$) plus regeneration for the "worthy condition," which may be expensive for very large systems.
2. Benchmark Diversity: While DevEval+ is structurally complex, the evaluation is limited to 20 modules, which might not fully capture the extreme edge cases of massive, multi-million line repositories.

---

> ### Author Rebuttal · Authors · 2026-03-31
>
> We thank the reviewer for the thoughtful and constructive review.
>
> ## Weakness 1: Computational Cost
> **End-to-end cost comparison (Table 1).** We conducted new experiments varying DocSearch's beam budget B to provide a comprehensive cost-performance comparison. At B=20, DocSearch costs \\$25.3, comparable to Claude Code+Topo+Test (\\$24.9), yet achieves φ̄=81.0%, a +9.5 pp advantage. At B=50 (\\$48.2), DocSearch delivers φ̄=93.3%, outperforming the strongest iterative baseline (60.9%) by over 32 pp at only 1.3× the cost.
>
> **Table 1: End-to-end cost (GPT-4o, DevEval+).**
>
> | Method | Cost (\\$) | φ̄ (%) |
> |---|---|---|
> | Iterative | 35.5 | 56.6 |
> | Iterative+Topo | 36.2 | 60.9 |
> | Cursor+Topo+Test | 13.8 | 59.4 |
> | Claude Code+Topo+Test | 24.9 | 71.5 |
> | DocSearch (B=20) | 25.3 | 81.0 |
> | DocSearch (B=30) | 32.9 | 87.0 |
> | DocSearch (B=50) | 48.2 | 93.3 |
>
> **Cost worthiness.** We acknowledge that DocSearch's beam search and worthy condition involve multiple generation cycles. However, a token-level analysis reveals that this apparent overhead actually yields *superior* efficiency. Table 2 tracks φ̄ as a function of cumulative output tokens: at just 20k tokens, DocSearch (68.3%) already exceeds Iterative's *final* performance at 200k tokens (56.6%), a 10× efficiency gap. The full token-budget curve is shown in [Figure 1](https://anonymous.4open.science/r/docsearch_icml_rebuttal-2BF6/token_efficiency.png). Moreover, DocSearch is a one-time offline optimization: the optimized documentation is reused across all downstream tasks without re-incurring the search cost.
>
> **Table 2: Token-efficiency milestones — φ̄ (%) (GPT-4o, DevEval+).**
>
> | Output Tokens (k) | Iterative | Iter+Topo | DocSearch |
> |---|---|---|---|
> | 20 | 43.1 | 45.3 | 68.3 |
> | 50 | 53.7 | 56.4 | 88.8 |
> | 200 | 56.6 | 60.9 | 93.3 |
>
> ## Weakness 2: Benchmark Scale
>
> DocSearch's architecture processes one module at a time. A large repository decomposes into individual modules, each with its own dependency DAG, exactly the unit DocSearch operates on. The workflow (generate documentation → produce code → test in original repository environment) applies regardless of repository scale.
>
> Inspired by the reviewer's question, we conducted new experiments on [SWE-Dev](https://arxiv.org/abs/2505.16975), a repo-level feature-implementation benchmark where agents build new functionality within existing codebases. We select 10 packages (67 instances, 35 unique source files). DocSearch+Source improves +14.8 pp pass and +11.7 pp solve over the PRD baseline (Table 3), confirming that DocSearch remains effective on repo-level benchmark.
>
> **Table 3: Feature implementation on SWE-Dev.**
> | Configuration | φ̄ (%) | \|S\| (%) |
> |---|---|---|
> | Source + PRD | 51.9 | 27.6 |
> | DocSearch + Source | 66.7 | 39.3 |
>
> ## Q1: Integration with Agent Skills
>
> DocSearch-optimized documentation can directly serve as a reusable tool output within multi-agent workflows: an agent calls DocSearch offline to optimize module documentation, then any downstream agent uses the optimized documentation for code generation without re-incurring the search cost. We validate this on two downstream tasks:
>
> **Task 1: Expanded cross-language transfer.** We expanded Python→Java from 10 to 16 modules (181 entities, 519 tests). Doc-only (φ̄=71.4%) outperforms Source-only (50.4%) by 21.0 pp; Doc+Source achieves 80.5%.
>
> **Task 2: New experiment on SWE-Dev (Table 3).** DocSearch+Source improves +14.8 pp pass and +11.7 pp solve over PRD baseline, demonstrating that optimized documentation is a more effective specification than hand-written PRDs even in a repo-level setting.
>
> **Generalization to skill search.** The DocSearch framework generalizes beyond documentation: the core loop (generate specification → execute task → diagnose failures → refine) is agnostic to whether the "specification" is documentation, a tool description, or a skill prompt. One could retain dependency-guided scheduling and the worthy condition while replacing code generation with any skill-executable task. This is an exciting direction we plan to explore, and we thank the reviewer for highlighting this connection.
>
> ## Q2: Scalability to Repository-Level Benchmarks
>
> DocSearch is feasible for repo-level tasks, its modular architecture processes each module independently regardless of repository size. To demonstrate repo-level applicability, we evaluate on SWE-Dev (Table 3), where agents implement new features requiring detailed understanding of existing module internals. For broader repo-level tasks like SWE-bench, DocSearch could serve as a complementary component: a retrieval system handles localization, and DocSearch optimizes documentation for the identified modules to enable faithful implementation. We will add this discussion to the revised paper.

---

> > ### Author Rebuttal · Reviewer_4z5r · 2026-04-04
> >
> > The authors' response has addressed my concerns.

---

> > > ### Author Response · Authors · 2026-04-04
> > >
> > > Thank you for the thoughtful review and for confirming that our responses have addressed your concerns. Your questions on computational cost and benchmark scalability prompted us to conduct the cost-performance analysis and SWE-Dev experiments, which we believe provide a more complete picture of DocSearch's practical strengths. We appreciate that your feedback helped us articulate these aspects more clearly.

---

### Decision · Program_Chairs · 2026-04-30

**Decision:**

Accept (regular)

**Comment:**

The paper studies the task of optimizing code documentation so that LLM agents can read them effectively, and frames it as a black-box bi-level search problem. The problem is well motivated and the proposed solution is conceptually novel and practically effective. The exposition is clear and the technical evaluation is sound; while reviewers were initially concerned about the cost, test-quality sensitivity and scalability, the authors adequately addressed these concerns in their rebuttal. During the discussion phase, the authors provided cost and token efficiency analyses, a paired human evaluation that showed improved documentation quality, repo-level SWE-Dev experiments, and clarifications on the implementation of the "worthy condition" that they proposed. Overall, the proposed method produces gains over strong baselines, avoids regression, has practical cost trade-offs that can have potentially large impact on an emerging task/challenge.